# The (Marginal) Value of a Search Ad: An Online Causal Framework for Repeated Second-price Auctions

**Yuxiao Wen** [1]  **Zihao Hu** [2]  **Yanjun Han** [3]  **Yuan Yao** [2]  **Zhengyuan Zhou** [4]

## Abstract

Existing auto-bidding algorithms in digital advertising often treat the value of an ad opportunity as the revenue obtained when an ad is shown and/or clicked, and bid accordingly. This can lead to wasteful spending because the true value is the marginal gain from paid exposure: even without winning a sponsored slot, an advertiser may still earn revenue via an organic search result (e.g., on Google or Amazon). Motivated by recent work, we model ad value as a treatment effect—the outcome difference between winning and losing the auction—and study online learning for bidding in second-price (Vickrey) auctions under this causal perspective. We develop algorithms that attain rate-optimal regret under several feedback models. A key ingredient exploits the information revealed by the second-price payment rule, which strictly improves regret relative to analogous learning problems in first-price auctions.

## 1. Introduction

Over the past years, advertisement has largely shifted from traditional promotions to digital advertising (Wagner, 2019). Online advertising platforms—such as Amazon, Google, and Meta—have access to rich customer information and can help advertisers better target the intended audience tailored to their brands. In many of these advertising platforms and in particular the sponsored advertisement in mainstream search engines, second-price auctions (SPAs) (Vickrey, 1961) are employed to sell the ad inventory for their truthful nature (Lucking-Reiley, 2000; Klemperer, 2018; Lucking-Reiley et al., 2007). In SPAs, the best practice of the bidder is to simply bid her own valuation of the item.[1]

In practice, however, advertisers face the crucial challenge of evaluating the actual value of the ad opportunity they are bidding on. This value is typically measured by the user's click-through rate (CTR) or conversion rate, which varies with the specific user or other environmental factors and is not known to the bidder. Note that an inaccurate valuation can lead to either overbidding or underbidding and thereby largely hurt the bidder's utility. To address this challenge and maximize the utility, one must rely on the rich contextual information to jointly estimate the ad value and make near-optimal bids on the fly (Weed et al., 2016; Feng et al., 2018; Cesa-Bianchi et al., 2024).

A key and yet vastly overlooked aspect in the literature is that the bidder's product may still be included in the organic search results and receive a click, even if the bidder has lost the auction for a sponsored slot in the user's search. In other words, the utility of losing an auction is *not necessarily zero*, and the value of an ad slot should be measured by the *marginal* gain as opposed to the sole outcome of winning the auction.

As a concrete motivation, suppose the user searches for cat food and looks for a few trusted brands he has purchased before. Such loyal users always ignore sponsored ads and head straight to the familiar brands in the search results. For those brands, while the outcome of serving ads is high, the marginal gain from serving ads is zero. Another motivation is that the brands can already be ranked top in the organic search results and receive a high CTR. They will gain little extra exposure by winning the auction and placing themselves among the sponsored slots. In either scenario, the marginal value of such an ad opportunity is small to the advertisers, but existing metrics mistake it to be a highly valuable one if value is measured only by the winning outcome.

To bridge this gap, a recent line of work proposes to model the marginal value as a *treatment effect*, that is the outcome difference between winning and losing the auction (Waisman et al., 2024; Wen et al., 2025a). Wen et al.

[1]Department of Computer Science, New York University [2]Department of Mathematics, Hong Kong University of Science and Technology [3]Center for Data Science, New York University [4]Stern School of Business, New York University. Correspondence to: Yuxiao Wen <yuxiaowen@nyu.edu>.

*Proceedings of the 43rd International Conference on Machine Learning*, Seoul, South Korea. PMLR 306, 2026. Copyright 2026 by the author(s).

---

[1]Throughout this work, we use advertisers and bidders interchangeably.

(2025a) studies this problem of jointly estimating treatment effect and bidding in first-price auctions (FPAs) and derives near-optimal algorithms under different feedback structures. Specifically, they consider two types of feedback on the highest other bid (HOB):

- *Full-information*: the bidder always observes the HOB at the end of auction.

- *Binary*: the bidder only observes the win-loss indicator.

In FPAs, the optimal regret scales with $\widetilde{\Theta}(\sqrt{dT})$ under full-information and $\widetilde{\Theta}_d(T^{\frac{2}{3}})$ under binary feedback (Wen et al., 2025a). Here $T$ denotes the horizon of the bidding process and $d$ the feature dimension. In this work, we focus on SPAs and provide a complete picture across different feedback. In particular, we prove that bidding in SPAs is fundamentally easier than in FPAs under binary feedback. Our contributions are detailed as follows:

- **(Problem formulation)** We introduce the formulation of treatment effect estimation to bidding in repeated SPAs. Crucially, we identify the information revealed by the payment in SPAs that is key to facilitating the learning process.

- **(Optimal regret)** By carefully exploiting the payment rule in SPAs, we establish an improved regret $\widetilde{\Theta}(\sqrt{dT})$ under the binary feedback, which outlines a fundamental difference between SPAs and FPAs. Together with a lower bound under full-information feedback, we provide a complete picture of regret characterizations for SPAs under both types of feedback (Table 1).

- **(Algorithmic generalization)** To circumvent unrealistic overlap conditions in treatment effect estimation, we improve upon the causal inference approach employed in Wen et al. (2025a). We relax the propensity score estimation condition to accommodate arbitrary estimation approaches, including the estimation entailed by our specific information structure. We also generalize the assumption on HOB distributions to incorporate point mass, handling a much broader class of distributions.

- **(Practical implementation)** For practical concerns, we also include an algorithm variant that discards complicated theoretical devices (Algorithm 1[†]). Its performance and insights are discussed in Section 5.

## 1.1. Related Work

**Bidding in repeated auctions**  Research in auction theory has a long history. Early work of auctions, particularly SPAs and FPAs, typically takes a game-theoretic perspective to understand the equilibria of the bidding behaviors (Vickrey, 1961; Myerson, 1981; Klemperer, 1999). A more recent line of research, also more relevant to this work, combines bidding in repeated auctions with the view of learning to address uncertainties faced by the bidder (Blum et al., 2004; Devanur & Kakade, 2009; Weed et al., 2016; Mohri & Medina, 2016; Feng et al., 2018; Han et al., 2020; 2025; Wen et al., 2025a; Hu et al., 2025). A majority of this literature assumes a known ad value and focuses on the uncertainty in the HOBs in FPAs, since the optimal bidding strategy in SPAs is trivial when the value is known. Nonetheless, learning HOBs turns out critical in our work because of the unknown valuation. As we model the value as a treatment effect and estimate it through the lens of causal inference, the knowledge on HOBs is required when computing the propensity score of the treatment (which is a successful display of the ad). In FPAs, Han et al. (2025) proposes the idea of interval splitting to handle HOB estimation under incomplete feedback. Building on this idea, we develop an estimated propensity score in SPAs with bid-dependent confidence width for value estimation.

**Estimating unknown ad value**  There is also a rich line of literature that addresses the uncertainty in ad value. Weed et al. (2016) takes a bandit approach and considers regret minimization in the setting where the bidder observes the ad value if she wins the SPA. This is generalized to FPAs by Feng et al. (2018) and Cesa-Bianchi et al. (2024). More recently, Waisman et al. (2024) proposes to model the ad value as a treatment effect, and Wen et al. (2025a) provides near-optimal algorithms for joint treatment effect estimation and bidding in FPAs. To highlight our contributions, we make a more detailed comparison with Wen et al. (2025a). We study the natural extension to SPAs, as a user can interact with both sponsored ads (the winning bidders) and organic results (the losing bidders) in a search, making treatment effect modeling a particular fit for search ads. Importantly, while the optimal regret scales with $\widetilde{\Theta}_d(T^{\frac{2}{3}})$ in the FPAs under binary feedback (Wen et al., 2025a), it is not tight in the SPAs. The winner in a SPA observes the HOB when she pays. This payment yields additional and asymmetric information that makes winning more profitable and improves the regret to $\widetilde{\Theta}(\sqrt{dT})$. To achieve this improvement, we generalize the condition required for HOB estimation in Wen et al. (2025a) to incorporate any form of error bounds, which may be of independent interest to future work.

**Incrementality/lift bidding**  Prior to this work, a line of growing literature also propose to model the ad value as a causal difference under the name of *incrementality* or *lift*, which indicates the increasing interest from practitioners. Nonetheless, most works focus on the empirical perspectives due to the difficulties in causal online learning (Gordon et al., 2019; Lewis & Wong, 2022; Gordon et al., 2023; Wais-

man et al., 2024). Others build theoretical insights upon often unrealistic assumptions, such as the access to massive randomized exploration data (Bompaire et al., 2021; Johnson et al., 2017; Xu et al., 2016) or the overlap condition (Badanidiyuru Varadaraja et al., 2022).[2] The theoretically near-optimal algorithms in FPAs, without restrictive conditions, are provided by the recent work Wen et al. (2025a).

**Linear contextual bandits** This work also closely relates to the literature on linear contextual bandits, as we will impose a linear model on the treatment effect in features. In the bandit literature, the learner always observes the value after choosing an arm. The optimal regret scales with $\widetilde{\Theta}(d\sqrt{T})$ when the arm space is continuous (Dani et al., 2008; Abbasi-Yadkori et al., 2011) and $\widetilde{\Theta}(\sqrt{dT})$ when the arm set is finite (Auer, 2002; Chu et al., 2011). While these results do not apply as the value is not observed in our case, their ideas shed light to understanding the convergence of value estimation in linear models, when coupled with appropriate causal inference techniques.

### 1.2. Notations

Let $[n] = \{1, 2, \ldots, n\}$ for positive integer $n$. We define the indicator function $\mathbb{1}[E]$ to be 1 if the event $E$ occurs and 0 otherwise. For vector $x \in \mathbb{R}^d$ and positive semi-definite (PSD) matrix $A \in \mathbb{R}^{d \times d}$, define $\|x\|_A = \sqrt{x^\top A x}$. We use standard asymptotic notations $O(\cdot), \Omega(\cdot)$, and $\Theta(\cdot)$ to suppress constant factors, and $\widetilde{O}(\cdot), \widetilde{\Omega}(\cdot)$, and $\widetilde{\Theta}(\cdot)$ to suppress poly-logarithmic. We write $\widetilde{\Theta}_d(\cdot)$ when polynomial factors in $d$ are also suppressed. For a random variable $X$, $\mathbb{E}[X]$ and $\mathrm{Var}(X)$ denote its mean and variance. For a cumulative distribution function (CDF) $G$ on $[0, 1]$, let its (pseudo-)inverse be $G^{-1}(z) = \inf\{x \in [0, 1] : G(x) \geq z\}$.

### 1.3. Organization

Section 2 summarizes the problem setup and our main results. Then we proceed with two estimation components. We explain the *interval splitting* idea that exploits the second-price payment for a finer-grid HOB estimation in Section 3. Section 4 lists the key challenges and insights in the causal estimation of the treatment ad value $\Delta v_t$. A simplified implementation from the practical perspective and its empirical validation is provided in Section 5.

## 2. Problem Formulation and Main Results

### 2.1. Problem Formulation

Consider a single bidder who jointly estimates the unknown (marginal) value and bids in repeated SPAs over a horizon of

---

length $T$. A feature vector $x_t \in \mathbb{R}^d$ is revealed to the bidder at the beginning of every auction, or every time $t \in [T]$. Then the bidder submits a bid $b_t$, while the HOB $m_t$ is drawn by nature. If $b_t \geq m_t$, the bidder wins the auction, pays the HOB $m_t$, and receives a winning outcome $v_{t,1}$; if $b_t < m_t$, the bidder loses, pays nothing, and receives a baseline outcome $v_{t,0}$. We consider *binary feedback* where the bidder observes no additional information other than the win-loss indicator. The observable (inferred from the auction outcome) is $v_{t,1}$ if won and $v_{t,0}$ if lost. Crucially, as opposed to FPAs, the bidder observes the HOB $m_t$ from the payment rule if and only if she wins, creating an asymmetric information structure that favors winning. The payoff function at time $t$ is

$$r_t(b) := \mathbb{1}[b \geq m_t](v_{t,1} - m_t) + \mathbb{1}[b < m_t]v_{t,0}$$
$$= \mathbb{1}[b \geq m_t](v_{t,1} - v_{t,0} - m_t) + v_{t,0}.$$

For simplicity, we assume $\|x_t\|_2 \leq 1$ and $v_{t,1}, v_{t,0}, m_t \in [0, 1]$. To address this problem, we consider a linear model for the ad value $\Delta v_t := v_{t,1} - v_{t,0}$. Note that this value $\Delta v_t$ is *never* observed, posing the need for a causal inference approach.

**Assumption 2.1** (Linear model). The treatment effect satisfies $\mathbb{E}[\Delta v_t] = \theta_*^\top x_t$ at every $t$ for some unknown parameter $\theta_* \in \mathbb{R}^d$ with $\|\theta_*\|_2 \leq 1$.

**Assumption 2.2** (Stochastic HOB). The HOB $m_t$ is drawn from an unknown i.i.d. distribution with a $(\omega, \lambda)$-locally-bounded CDF $G$ for some known constants $\omega, \lambda \in (0, 1)$.

**Assumption 2.3** (Oblivious context). Conditioned on contexts $\{x_t\}_{t \in [T]}$, the values $\{(v_{t,1}, v_{t,0}, m_t)\}_{t \in [T]}$ are independent over time.

**Definition 2.4.** A CDF $G$ is called $(\omega, \lambda)$-locally-bounded if for every $b_1, b_2 \in [0, 1]$, $|b_1 - b_2| \leq \omega$ implies $|G(b_1) - G(b_2)| \leq \lambda$.

The reason behind the i.i.d. HOB model is that the population of competing bidders is typically large in ad exchanges and relatively stationary over time.[3] So we expect the competing bid $m_t$ to average out and have a stationary behavior. Stationarity has been imposed in prior literature and in practice (Mohri & Medina, 2016; Han et al., 2025). Assumption 2.3 simply states that the context sequence is oblivious to the realized history, which is also standard in the literature and subsumes the case when $x_t$ is i.i.d. (Auer, 2002; Abbasi-Yadkori et al., 2011).

Let $g$ and $G$ denote the HOB density and CDF, respectively. The expected payoff is

$$\overline{r}_t(b) := G(b)\theta_*^\top x_t - \int_0^b g(m)m\,\mathrm{d}m + \mathbb{E}[v_{t,0}]. \quad (1)$$

---

To measure the performance of a bidding algorithm $\pi$, we consider the following notion of regret. It competes against the hindsight oracle that perfectly knows $\theta_*$ and $G$.

$$R(\pi) := \mathbb{E}\left[\sum_{t=1}^{T} \max_{b_t^* \in [0,1]} \overline{r}_t(b_t^*) - \overline{r}_t(b_t)\right] \quad (2)$$

where the bid sequence $(b_t)_t$ is selected by the algorithm $\pi$ and the expectation is taken over any randomness in the algorithm and the bidding process.

### 2.1.1. EXAMPLE HOBS

To be concrete, we list a few examples of HOB distributions with $(\omega, \lambda)$-locally-bounded CDFs.

**Continuous distribution with bounded density.** Suppose the HOB density is upper bounded by $U > 0$, which implies that the CDF $G$ is $U$-Lipschitz. Then we have $\lambda = U\omega$ for any $\omega \in (0, 1)$.

**Distribution with atoms.** An atom or a point mass refers to a value $m \in [0, 1]$ such that $\mathbb{P}(m_t = m) > 0$. The HOB distribution is allowed to have atoms as long as, over any interval $[a, a + \omega] \subseteq [0, 1]$ of length $\omega$, the probability $\mathbb{P}(a < m_t \leq a + \omega) = G(a + \omega) - G(a) \leq \lambda$ is bounded.

### 2.2. Main Results

The main theorem of this work provides a near-optimal regret guarantee for the proposed bidding algorithm.

**Theorem 2.5.** *Suppose Assumption 2.1–2.3 hold. Under this binary feedback, there is a bidding algorithm $\pi$ that achieves*

$$R(\pi) = O(\sqrt{dT} \log^3 T + d \log^5 T).$$

To complement our regret bound, we present a matching minimax lower bound. Let the minimax regret be

$$R^* = \inf_{\pi} \sup_{G, \theta_*} R(\pi; G, \theta_*)$$

where the sup is taken over any pair of parameters $(G, \theta_*)$ that satisfies Assumption 2.1–2.3, and $R(\pi; G, \theta_*)$ denotes the regret of algorithm $\pi$ under the corresponding problem instance. The next result indicates that the algorithm in Theorem 2.5 is optimal up to poly-logarithmic factors in the minimax sense.

**Theorem 2.6.** *Suppose Assumption 2.1–2.3 hold. Even if the CDF $G$ is perfectly known, when $T \geq d^2$, it holds that*

$$R^* = \Omega(\sqrt{dT}).$$

Since Theorem 2.6 holds even under full-information HOB feedback, we have tight regret under both full-information and binary feedback, as summarized in Table 1.

*Table 1.* Optimal Regret in SPAs under Two HOB Feedbacks

| | Full-information | Binary |
|---|---|---|
| Upper Bound (Thm. 2.5) | $\widetilde{O}(\sqrt{dT})$ | $\widetilde{O}(\sqrt{dT})$ |
| Lower Bound (Thm. 2.6) | $\Omega(\sqrt{dT})$ | $\Omega(\sqrt{dT})$ |

## 3. HOB Estimation from Payment

In this section, we formalize how to learn the HOB distribution from the second-price payment rule, which is informative only when the bidder wins the auction. Throughout the remaining work, we consider the discretized bids

$$\mathcal{B} := \{b^j : j = 1, 2, \ldots, \lceil \sqrt{T} \rceil\}, \quad b^j = \frac{j-1}{\sqrt{T}}. \quad (3)$$

### 3.1. One-sided Feedback and More

Note that a higher bid reveals more information about the HOB CDF $G$ under the second-price payment rule. Indeed, for $b < b'$, one can always infer $\mathbb{1}[b \geq m_t]$ from $\mathbb{1}[b' \geq m_t]$: if $\mathbb{1}[b' \geq m_t] = 0$, then so is $\mathbb{1}[b \geq m_t] = 0$. Otherwise, since the bidder wins the auction and pays the HOB, the bidder knows $m_t$ and can compute $\mathbb{1}[b \geq m_t]$. This gives the name one-sided feedback. Nonetheless, as studied in the bandit literature, this one-sided feedback alone is insufficient to achieve the optimal regret $\widetilde{O}(\sqrt{T})$ when the sequence of realized values $(\Delta v_t)_t$ is unconstrained (Wen et al., 2024; Han et al., 2025).

A key observation from Han et al. (2025) is that the one-bit feedback $\mathbb{1}[b \geq m_t]$ from a lower bid $b < b'$ also provides *partial* information to the CDF $G(b')$ of a higher bid. Indeed, since

$$G(b') = \mathbb{E}[\mathbb{1}[b' \geq m_t]] = \mathbb{E}[\mathbb{1}[b \geq m_t] + \mathbb{1}[b' \geq m_t > b]]$$
$$= G(b) + \mathbb{P}(b' \geq m_t > b),$$

observations from bidding $b$ help estimate the first part $G(b)$. To utilize this additional information, for each discretized bid $b^j \in \mathcal{B}$ in (3), we have

$$G(b^j) = \sum_{i \leq j} p^i$$

where $p^i := \mathbb{P}(b^i \geq m_t > b^{i-1})$. Then we estimate each $p^i$ individually using historical observations. This is beneficial for two reasons.

First, as commented earlier, observations from higher bids imply observations on lower bids. Therefore, as described in Figure 1, more data is available for smaller bids. By estimating each $p^i$ separately for $i \leq j$, we can derive a tighter confidence width for $G(b^j)$, which is crucial to improve the regret dependence from $T^{\frac{2}{3}}$ to $\sqrt{T}$.

Second, an estimator $\widehat{p}^i$ converges faster to the truth when the target probability $p^i$ is smaller than a constant level. By Bernstein's concentration (Lemma K.1), this value $p^i = o(1)$ shows up in the confidence width and enables fast learning. Nonetheless, since $p^i$ is unknown, this tight confidence width cannot be computed directly. We adopt the solution by Han et al. (2025) to use $O(\sqrt{T})$ initial samples to compute an initial estimator $\widehat{p}_0^i$ for each $i$ in Algorithm 3 and prove that it suffices for our target regret.

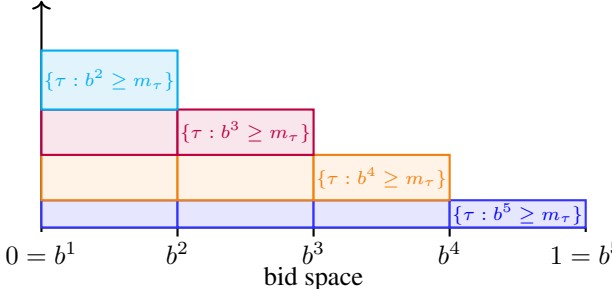

*Figure 1.* **Illustration of the one-sided feedback inferred from payment.** This is a toy example with discretization size $|\mathcal{B}| = 5$. Because of the second-price payment, the bidder can infer $\mathbb{1}[b^i \geq m_\tau]$ from $\mathbb{1}[b^j \geq m_\tau]$ whenever $b^i \leq b^j$. Consequently, the smaller bid intervals always have more observations.

## 3.2. HOB Estimation

To elaborate on the idea above, we consider the estimation of $\{G(b) : b \in \mathcal{B}\}$ at a given time $t$. Suppose we have a subset of time indices $\Phi_t \subseteq [t-1]$ such that the HOB observations $\{\mathbb{1}[b_\tau \geq m_\tau]m_\tau\}_{\tau \in \Phi_t}$ are *mutually independent* conditioned on $\{(b_\tau, x_\tau)\}_{\tau \in \Phi_t}$. For each bid index $j \in [\lceil\sqrt{T}\rceil]$, we can define the probability estimator

$$\widehat{p}_t^j := \frac{\sum_{\tau \in \Phi_t} \mathbb{1}[b_\tau \geq b^j]\mathbb{1}[b^j < m_\tau \leq b^{j+1}]}{\sum_{\tau \in \Phi_t} \mathbb{1}[b_\tau \geq b^j]}$$

and naturally the CDF estimator

$$\widehat{G}_t(b^j) := \sum_{i \leq j} \widehat{p}_t^i. \tag{4}$$

To derive an estimator for the reward in (1) later, note that the *expected payment* can be approximated as $\int_0^{b^j} g(m)m\mathrm{d}m = b^j G(b^j) - \int_0^{b^j} G(m)\mathrm{d}m \approx b^j G(b^j) - \frac{1}{\sqrt{T}}\sum_{i \leq j} G(b^i)$. It turns out critical to develop good estimators for both the CDF value and its discretized integral:

**Lemma 3.1.** *At each time $t \in [T]$ and given indices $\Phi_t \subseteq [t-1]$, suppose $\{\mathbb{1}[b_\tau \geq m_\tau]m_\tau\}_{\tau \in \Phi_t}$ are mutually independent conditioned on $\{(b_\tau, x_\tau)\}_{\tau \in \Phi_t}$. Let $n_t^j := \sum_{\tau \in \Phi_t} \mathbb{1}[b_\tau \geq b_j]$ for each index $j \in [\lceil\sqrt{T}\rceil]$. With probability at least $1 - T^{-3}$, it holds that*

$$|G(b^j) - \widehat{G}_t(b^j)| \leq u_t(b^j) \text{ and } \frac{1}{\sqrt{T}}\left|\sum_{i \leq j} G(b^i) - \widehat{G}_t(b^i)\right| \leq u_t(b^j)$$

*for every $j \in [\lceil\sqrt{T}\rceil]$ and $\widehat{G}_t$ defined in (4). Here*

$$u_t(b^j) := 8\sqrt{\sum_{k \leq j} \frac{2\log T}{n_t^k}\left(\widehat{p}_0^k + \frac{12\log T}{\sqrt{T}}\right) + \frac{8\log T}{n_t^j}}$$

*where $\widehat{p}_0^k$ estimates $p^k$ with at least $\lceil\sqrt{T}\rceil$ i.i.d. samples as defined in (8).*

The conditional independence required in Lemma 3.1 does not trivially hold and is addressed later in Section 4.4. For the purpose of demonstration, we shall assume it holds until Section 4.4.

# 4. Bidding with Unreliable Value Estimation

We now consider estimation of the linear parameter $\theta_*$ and the actual bidding algorithm. The estimation is "unreliable" in the sense that, as we will see shortly, the estimation error bound is in general *unbounded*. Indeed, because the value is a treatment effect, we would need observations of both winning and baseline outcomes $(v_{t,1}, v_{t,0})$ to accurately estimate $\theta_*$. Developing a good estimator $\widehat{\theta}_t$ is typically done via randomized experiments or guaranteed observations of both sides in causal inference (i.e. the overlap condition). However, as the bidder minimizes the regret, she is inclined to win the auctions with large $\Delta v_t$ and lose the ones with small $\Delta v_t$. Naively running randomized experiments would result in a suboptimal regret.[4]

To arrive at the optimal regret, we will give up on developing a reliable estimator $\widehat{\theta}_t$ for $\theta_*$ at any time $t$ during the bidding process. Instead, the performance of our estimator $\widehat{\theta}_t$ will depend on the bidding trajectory and have a generally unbounded error. Crucially, this large estimation error is neutralized by a decision step that carefully exploits the structure of the repeated auctions.

## 4.1. Inverse-propensity-weighted (IPW) Estimator

The value estimation starts with a standard causal inference concept: the IPW estimator. Since the treatment in our formulation corresponds to the display of an ad, the propensity score of bidding $b$ at time $t$ is $\mathbb{P}(b \geq m_t) = G(b)$. When $G$ is known, it is natural to consider the unbiased IPW estimator for $\Delta v_t$:

$$\widehat{e}_t(b) := \frac{\mathbb{1}[b \geq m_t]v_{t,1}}{G(b)} - \frac{\mathbb{1}[b < m_t]v_{t,0}}{1 - G(b)}.$$

While $G$ is not known, the bidder may maintain an estimator $\widehat{G}_t$ at each time $t$ and devise a variant of this IPW estimator.

---

[4]Suppose one uses $N$ randomized experiments to draw bids $b_t \sim \text{Unif}\{0, 1\}$, estimate $\theta_*$ via these observations, and then commit to the estimator. Trading off $N$ leads to regret $\widetilde{O}(T^{\frac{2}{3}})$.

This work considers a simple alternative:

$$\widetilde{e}_t(b) := \frac{\mathbb{1}[b \geq m_t]v_{t,1}}{\widehat{G}_t(b)} - \frac{\mathbb{1}[b < m_t]v_{t,0}}{1 - \widehat{G}_t(b)}. \tag{5}$$

The performance of this estimator is summarized by the following result. Note that it by no means bounds the bias nor the variance of the IPW estimator in (5), as $\sigma_t(b)$ can go to infinity when $\widehat{G}_t(b)$ is close to 0 or 1. The purpose is to find computable proxies for its bias and variance in further variance reduction steps.

**Lemma 4.1** (Bias-variance proxy)**.** *At time $t$, suppose* $|\widehat{G}_t(b) - G(b)| \leq u_t(b)$ *for every bid $b \in [0,1]$ for some width $u_t(b)$. Then there exists a constant $c_0 = 4$ such that*

$$\begin{cases} \left|\mathbb{E}[\widetilde{e}_t(b)] - \theta_*^\top x_t\right| \leq c_0 \cdot u_t(b)\sigma_t(b) \\ \mathrm{Var}(\widetilde{e}_t(b)) \leq c_0 \cdot \sigma_t(b)^2 \end{cases}$$

*where*

$$\sigma_t(b) := \frac{1}{\widehat{G}_t(b)(1 - \widehat{G}_t(b))}. \tag{6}$$

*Remark* 4.2 (IPW design). In comparison, to arrive at an algorithm with no explicit bid experiments, Wen et al. (2025a) proposes a "truncated" IPW variant when the HOB estimator satisfies $|G(b) - \widehat{G}_t(b)| \lesssim \sqrt{G(b)(1 - G(b))/t} + 1/t$. This is a Bernstein-type error bound that requires the HOB estimation to be more accurate at the regime where the variance $G(b)(1 - G(b))$ is small. Clearly, as our estimation in (4) takes a piece-wise approach, the guarantees in Lemma 3.1 do not align with such a condition. By contrast, our simple IPW design and performance guarantees in Lemma 4.1 allow for an arbitrary HOB error $|\widehat{G}_t(b) - G(b)| \leq u_t(b)$. It remains open whether one can achieve no-experiment with a more delicate IPW design in our setting.

Given the computable variance proxy $\sigma_t(b)^2$ of the IPW estimator in Lemma 4.1, we solve the weighted least squares to find an estimator for the linear parameter $\theta_*$. Specifically, given a selected subset of time indices $\Phi_t \subseteq [t-1]$ at time $t$, consider

$$\widehat{\theta}_t = \arg\min_{\theta \in \mathbb{R}^d} \sum_{\tau \in \Phi_t} \sigma_\tau(b_\tau)^{-2}(\widetilde{e}_\tau(b_\tau) - \theta^\top x_\tau)^2 + \|\theta\|_2^2. \tag{7}$$

The weights $\sigma_\tau(b_\tau)^{-2}$ are taken to address the heteroskedasticity in the constructed IPW estimators $\{\widetilde{e}_\tau(b_\tau)\}_{\tau \in \Phi_t}$, and a standard Ridge regularization is imposed. The solution to (7) takes the closed form as in Line 5 of Algorithm 1. The following result provides an error bound on the estimated treatment effect by the estimator defined in (7).

**Lemma 4.3.** *Suppose* $(v_{\tau,1}, v_{\tau,0})_{\tau \in \Phi_t}$ *are conditionally independent given* $(x_\tau, b_\tau, m_\tau)_{\tau \in \Phi_t}$, *and* $|G(b_\tau) - \widehat{G}_\tau(b_\tau)| \leq u_\tau$ *for every $\tau \in \Phi_t$. Then with probability at least $1 - T^{-3}$, it holds that*

$$|\widehat{\theta}_t^\top x_t - \theta_*^\top x_t| \leq \gamma\|x_t\|_{A_t^{-1}},$$

---

**Algorithm 1:** UCB Computation Routine

1 **Input:** Time indices $\Phi_t \subseteq [t-1]$, bid subset $B_t \subseteq \mathcal{B}$, CDF estimator $\widehat{G}_t$, historic widths $\{u_\tau\}_{\tau \in \Phi_t}$ and width function $u_t$.

2 $\sigma_\tau \leftarrow \sigma_\tau(b_\tau)$ in (12) for $\tau \in \Phi_t$;

3 $A_t \leftarrow I + \sum_{\tau \in \Phi_t} \sigma_\tau^{-2} x_\tau x_\tau^\top$;

4 $z_t \leftarrow \sum_{\tau \in \Phi_t} \sigma_\tau^{-2} x_\tau \widetilde{e}_\tau$ where $\widetilde{e}_\tau = \widetilde{e}_\tau(b_\tau)$ as in (11);

5 Compute estimator $\widehat{\theta}_t \leftarrow A_t^{-1} z_t$.

6 Set $\gamma \leftarrow 1 + 14 \log T + 4\sqrt{\sum_{\tau \in \Phi_t} u_\tau^2}$.

7 **for** $b^j \in B_t$ **do**

8     Compute the following quantities with:

9     $w_{t,0}(b^j) \leftarrow \frac{8}{1-\lambda}(\widehat{G}_t(b^j)\gamma\|x_t\|_{A_t^{-1}} + 4u_t(b^j) + \frac{2}{\sqrt{T}})$.

10     $w_{t,1}(b^j) \leftarrow \frac{8}{1-\lambda}((1 - \widehat{G}_t(b^j))\gamma\|x_t\|_{A_t^{-1}} + 4u_t(b^j) + \frac{2}{\sqrt{T}})$.

11     $\widehat{r}_{t,0}(b^j) \leftarrow \widehat{G}_t(b^j)(\widehat{\theta}_t^\top x_t - b^j) + \sum_{i \leq j} \widehat{G}_t(b^i)$.

12     $\widehat{r}_{t,1}(b^j) \leftarrow \widehat{G}_t(b^j)(\widehat{\theta}_t^\top x_t - b^j) + \sum_{i \leq j} \widehat{G}_t(b^i) - \widehat{\theta}_t^\top x_t$.

13 **end**

14 Invoke Algorithm 2 with perturbation $c = \frac{\omega}{4}$, margin constant $\epsilon = \frac{1-\lambda}{8}$, CDF $\widehat{G}_t$, and value $\widehat{\theta}_t^\top x_t$ to get $[b_\mathrm{L}, b_\mathrm{R}]$ and index $q \in \{0, 1\}$.

---

*where $\gamma$ and $A_t$ are defined in Line 6 and Line 3 of Algorithm 1.*

We remark that the bound in Lemma 4.3 is in general *unbounded*. When the variances $\sigma_\tau$ are large in the definition of $A_t$ in Algorithm 1, this matrix norm $\|x_t\|_{A_t^{-1}}$ can be arbitrarily close to 1, i.e. being trivial. In other words, our estimation of the treatment effect $\theta_*^\top x_t$ can be arbitrarily bad or even vacuous.

**4.2. Neutralizing Bad Value Estimations**

We are now in the position to present a key decision step that *neutralizes* this potentially unbounded estimation error, named "the better of two upper confidence bounds (UCBs)" by Wen et al. (2025a). First, recall that maximizing the reward $\overline{r}_t$ in (1) is equivalent to maximizing any of the following two formulations:

$$\overline{r}_{t,0}(b) := G(b)(\theta_*^\top x_t - b) + \int_0^b G(m)\mathrm{d}m$$
$$= \overline{r}_t(b) - \mathbb{E}[v_{t,0}]$$
$$\overline{r}_{t,1}(b) := -(1 - G(b))\theta_*^\top x_t - G(b)b + \int_0^b G(m)\mathrm{d}m$$
$$= \overline{r}_t(b) - \mathbb{E}[v_{t,1}].$$

Crucially, they have different dependence on the value $\theta_*^\top x_t$, which motivate the definition of the plug-in estimators $\widehat{r}_{t,0}$ and $\widehat{r}_{t,1}$ in Algorithm 1. It is rather straightforward to show that for every $b \in [0,1]$,

$$|\overline{r}_{t,0}(b) - \widehat{r}_{t,0}(b)| \leq w_{t,0}(b), \quad |\overline{r}_{t,1}(b) - \widehat{r}_{t,1}(b)| \leq w_{t,1}(b);$$

detailed computations are deferred to the proof of Lemma 4.4 in Appendix F.

Now there are two pairs of (reward estimator, confidence width) we can optimize over to select the final bid $b_t$. Suppose we apply the classical UCB algorithm on the formulation $\overline{r}_{t,0}$, such that $b_t = \arg\max_b \widehat{r}_{t,0}(b) + w_{t,0}(b)$. Standard analysis yields a bound on the instantaneous regret $\overline{r}_{t,0}(b_t^*) - \overline{r}_{t,0}(b_t) \lesssim w_{t,0}(b_t)$ where $b_t^*$ is the hindsight optimal bid. Now, if $w_{t,0}(b_t) \leq w_{t,1}(b_t)$, we have chosen the better formulation: the instantaneous regret scales with

$$\min\{w_{t,0}(b_t), w_{t,1}(b_t)\} \propto \widehat{G}_t(b_t)(1-\widehat{G}_t(b_t)) = \sigma_t(b_t)^{-1}.$$

Recall that the value estimation error in Lemma 4.3 is large when the variance $\sigma_t$ is large, which surprisingly implies a vanishing regret under the better UCB formulation.

But what if $w_{t,0}(b_t) > w_{t,1}(b_t)$? After all, we did not know the selected $b_t$ from $\overline{r}_{t,0}$ leads to a small $w_{t,0}(b_t)$ at the time we chose to use the UCB of $\overline{r}_{t,0}$ over $\overline{r}_{t,1}$.

### 4.3. UCB Formulation Selection

It is not straightforward to select the better UCB formulation, since we do not know $b_t$ beforehand. The selection is handled by Algorithm 2. At a high level, it prunes the given bid subset $B_t \subseteq \mathcal{B}$ at a coarse level such that the CDF values $\widehat{G}_t(b)$ of all remaining bids are either close to 0 or 1. When they are close to 0, it is clear that $w_{t,0}(b) \leq w_{t,1}(b)$ up to constants, and hence we favor $\widehat{r}_{t,0}$ over $\widehat{r}_{t,1}$; vice versa. This is formalized by the following result:

**Lemma 4.4** (Small width for selected UCB). *Suppose the events in Lemma 3.1 and 4.3 hold. Also suppose $|G(b^j) - \widehat{G}_t(b^j)| \leq u_t(b^j)$ and $\frac{1}{\sqrt{T}}\left|\sum_{i \leq j} G(b^i) - \sum_{i \leq j} \widehat{G}_t(b^i)\right| \leq u_t(b^j)$ for each $b^j \in \mathcal{B}$ for the given width function $u_t$. For the index $q \in \{0,1\}$ returned by Algorithm 2 with constant perturbation $c = \frac{\omega}{4}$, when $\sup_{b \in \mathcal{B}, q \in \{0,1\}} \frac{1-\lambda}{8} \cdot w_{t,q}(b) \leq C(\lambda, \omega)$,[5] it holds that*

$$|\overline{r}_{t,q}(b) - \widehat{r}_{t,q}(b)| \leq \frac{1-\lambda}{8} \cdot w_{t,q}(b) \leq \min\{w_{t,0}(b), w_{t,1}(b)\}$$

*for all $b \in [b_L, b_R]$ in Algorithm 1, and $\arg\max_{b \in \mathcal{B}} \overline{r}_t(b) \in [b_L, b_R]$.*

The claims in Lemma 4.4 show that, when the estimation errors are smaller than a constant, the index $q$ by Algorithm 2 indeed finds the better UCB for us, and the optimal bid (in the discretized space) lies in the returned interval $[b_L, b_R]$. The latter allows us to focus our bidding process only within $[b_L, b_R]$, over which we *know* the better UCB.

---

[5]The constant $C(\lambda, \omega)$ only depends on $\lambda$ and $\omega$ and is defined explicitly in Appendix F.

---

**Algorithm 2:** UCB Selection

1 **Input:** perturbation $c > 0$, margin $\epsilon > 0$, estimated CDF $\widehat{G}_t$, estimated value $\widehat{v}$.

2 Compute
$$b_+ \leftarrow \arg\max_{b^j \in \mathcal{B}} \widehat{G}_t(b^j)(\widehat{v} + c) - \frac{1}{\sqrt{T}}\sum_{i \leq j} \widehat{G}_t(b^i)$$

3 and
$$b_- \leftarrow \arg\max_{b^j \in \mathcal{B}} \widehat{G}_t(b^j)(\widehat{v} - c) - \frac{1}{\sqrt{T}}\sum_{i \leq j} \widehat{G}_t(b^i).$$

4 Set $S \leftarrow \{b \in \mathcal{B} : \widehat{G}_t(b_-) - c \leq \widehat{G}_t(b) \leq \widehat{G}_t(b_+) + c\}$

5 Set $b_L \leftarrow \min S$ and $b_R \leftarrow \max S$.

6 Output interval $[b_L, b_R]$ and index $q = \mathbb{1}[\widehat{G}_t(b_L) \geq \epsilon]$.

---

### 4.4. Maintaining Conditional Independence

Recall that our estimation results, Lemmata 3.1 and 4.3, require conditional independence among the observations. This is not trivially guaranteed. For example, consider the HOB observations $\{\mathbb{1}[b_\tau \geq m_\tau]m_\tau\}_{\tau \in \Phi_t}$ at time $t$ for a subset $\Phi_t \subseteq [t-1]$. If we naively use all observations by taking $\Phi_t = [t-1]$, the bid $b_t$ we choose will depend on the information of $\{\mathbb{1}[b_\tau \geq m_\tau]m_\tau\}_{\tau \in [t-1]}$. Then when we condition on $(b_t, x_t)$ in the next time $t+1$ with $\Phi_{t+1} = [t]$, the conditional independence breaks, as the new observation $\mathbb{1}[b_t \geq m_t]m_t$ depends on the previous information of $\{\mathbb{1}[b_\tau \geq m_\tau]m_\tau\}_{\tau \in [t-1]}$ through $b_t$. A similar issue persists for the value observations in Lemma 4.3.

To address this issue, we adopt a master routine that partitions the history $[t-1]$ into $L = O(\log T)$ levels and, crucially, does not rely on the observations $\{\mathbb{1}[b_\tau \geq m_\tau]m_\tau\}_{\tau \in \Phi_t^{(\ell)}}$ at the current $\ell$-th level to make the decision $b_t$. This solution is standard: it was first proposed in the bandit literature (Auer, 2002) and has been widely applied in similar learning problems (Chu et al., 2011; Han et al., 2025; Wen et al., 2026).

To give an intuition, Algorithm 3 loops through the levels $\ell = 1, 2, \ldots, L$ at each time $t$. At each level $\ell$, it computes the confidence width of the reward estimators (in our case, the reward $\widehat{r}_{t,q}$ for the selected UCB $q \in \{0,1\}$), which crucially does not depend on the realized HOB and value observations. Indeed, the widths $w_{t,0}$ and $w_{t,1}$ defined in Algorithm 1 use only the number of observations and the contexts $(x_\tau)_{\tau \in \Phi_t}$ for the given subset $\Phi_t \subseteq [t-1]$. This bypasses the complicated dependence structure mentioned above. The time index $t$ is assigned to the $\ell$-th level if the confidence widths of the reward estimators lie in $[2^{-\ell}, 2^{-(\ell-1)})$ at a high level, by comparing the computed widths with this prespecified level of accuracy.

For the ease of notation, let $b_t = b^{j_t} \in \mathcal{B}$ with index $j_t$ denote the selected discretized bid, where $\mathcal{B}$ is defined in (3). To distinguish the initialization phase $[(L+1)T_0]$ and the remaining times in Algorithm 3, we define the initial

**Algorithm 3:** A Master Routine

1. **Input:** Time horizon $T$, HOB parameters $(\omega, \lambda)$.
2. **Initialize:** set $L = \lceil \log \sqrt{T} \rceil$, $J = \lceil \sqrt{T} \rceil$, and discretization $\mathcal{B}$ as in (3). Set $\Phi_1^{(\ell)} \leftarrow \varnothing$ for $\ell \in [L]$.
3. Set $T_0 \leftarrow \lceil \sqrt{T} \log T \rceil$.
4. **for** *time* $t = 1$ **to** $(L+1) \cdot T_0$ **do**
5.      Bid $b_t \leftarrow 1$ and observe $m_t$.
6. **end**
7. **for** *time* $t = (L+1) \cdot T_0 + 1$ **to** $T$ **do**
8.      Observe the context vector $x_t \in \mathbb{R}^d$.
9.      Initialize $B_1 \leftarrow \mathcal{B}$.
10.      **for** *level* $\ell = 1$ **to** $L$ **do**
11.          Denote $u_\tau^{(\ell)} \leftarrow u_\tau^{(\ell)}(b^{j_\tau})$ as in (9).
12.          Invoke Algorithm 1 with indices $\Phi_t^{(\ell)}$, space $B_\ell$, $\{u_\tau^{(\ell)}\}_{\tau \in \Phi_t^{(\ell)}}$, CDF $\widehat{G}_t^{(\ell)}$ as in (4), and $u_t^{(\ell)}$ to compute rewards $\{\widehat{r}_{t,q}^{(\ell)}(b)\}_{b \in B_\ell}$, widths $\{w_{t,q}^{(\ell)}(b)\}_{b \in B_\ell}$, UCB index $q \in \{0, 1\}$, and interval $\mathcal{I}_t^{(\ell)}$.
13.          **if** $\max_{b \in B_\ell, q \in \{0,1\}} w_{t,q}^{(\ell)}(b) > C(\lambda, \omega)$ **then**
14.              Uniformly randomly select bid $b_t = b^1, b^J$.
15.              Update: $\Phi_{t+1}^{(\ell)} \leftarrow \Phi_t^{(\ell)} \cup \{t\}$ and $\Phi_{t+1}^{(\ell')} \leftarrow \Phi_t^{(\ell')}$ for $\ell' \neq \ell$. Break.
16.          **else if** $\exists b \in B_\ell$ such that $w_{t,q}^{(\ell)}(b) > 2^{-\ell}$ **then**
17.              Choose this $b_t \leftarrow b$.
18.              Update: $\Phi_{t+1}^{(\ell)} \leftarrow \Phi_t^{(\ell)} \cup \{t\}$ and $\Phi_{t+1}^{(\ell')} \leftarrow \Phi_t^{(\ell')}$ for $\ell' \neq \ell$. Break.
19.          **else if** $w_{t,q}^{(\ell)}(b) \leq \frac{1}{\sqrt{T}}$ for all $b \in B_\ell$ **then**
20.              Choose $b_t \leftarrow \arg\max_{b \in B_\ell} \widehat{r}_{t,q}^{(\ell)}(b)$.
21.              Do not update: $\Phi_{t+1}^{(\ell')} \leftarrow \Phi_t^{(\ell')}$ for all $\ell' \in [N]$. Break.
22.          **else**
23.              We have $w_{t,q}^{(\ell)}(b) \leq 2^{-\ell}$ for all $b \in B_\ell$.
24.              Eliminate bids: $B_{\ell+1} \leftarrow$ $\left\{ b \in B_\ell : \widehat{r}_{t,q}^{(\ell)}(b) \geq \max_{b' \in B_\ell} \widehat{r}_{t,q}^{(\ell)}(b') - 2 \cdot 2^{-\ell} \right\} \cap \mathcal{I}_t^{(\ell)}$.
25.          **end**
26.      **end**
27.      Observe outcomes $\mathbb{1}[b_t \geq m_t]v_{t,1}, \mathbb{1}[b_t < m_t]v_{t,0}$, and payment $\mathbb{1}[b_t \geq m_t]m_t$.
28. **end**

times for each level as

$$\widehat{\Phi}_0^{(\ell)} = \{\ell T_0 + 1, \dots (\ell+1)T_0\}$$

and use $T_0$ held-out samples to derive initial probability

estimators: for each $j \in [\lceil \sqrt{T} \rceil]$, set

$$\widehat{p}_0^j := \frac{1}{T_0} \sum_{t \in \widehat{\Phi}_0^{(0)}} \mathbb{1}[b^j < m_t \leq b^{j+1}]. \qquad (8)$$

Then following Lemma 3.1, we define the CDF width in Algorithm 3 as

$$u_t^{(\ell)}(b^j) := 8\sqrt{\sum_{k \leq j} \frac{2 \log T}{n_{t,\ell}^k}\left(\widehat{p}_0^k + \frac{12 \log T}{\sqrt{T}}\right)} + \frac{8 \log T}{n_{t,\ell}^j}. \qquad (9)$$

where the number of observations is defined as

$$n_{t,\ell}^j := \sum_{\tau \in \Phi_t^{(\ell)} \cup \widehat{\Phi}_0^{(\ell)}} \mathbb{1}[b_\tau \geq b^j] \qquad (10)$$

for $b^j$ at the $\ell$-th level, where $\Phi_t^{(\ell)}$ is the subset of time indices associated with level $\ell \in [L]$.

**Lemma 4.5** (Lemma 14 of (Auer, 2002)). *Under Assumption 2.3, for every level $\ell \in [L]$ and time $t \in [T]$ in Algorithm 3, $(\mathbb{1}[b_\tau \geq m_\tau]m_\tau)_{\tau \in \Phi_t^{(\ell)} \cup \widehat{\Phi}_0^{(\ell)}}$ are conditionally independent given $(x_\tau, b_\tau)_{\tau \in \Phi_t^{(\ell)}}$, and $(v_{\tau,1}, v_{\tau,0})_{\tau \in \Phi_t^{(\ell)}}$ are conditionally independent given $(x_\tau, b_\tau, m_\tau)_{\tau \in \Phi_t^{(\ell)}}$.*

### 4.5. Random Exploration on the Fly

The final piece of Algorithm 3 is the random bid selection in Line 13. This exploration is to satisfy the condition in Lemma 4.4. In particular, if the confidence width is larger than $C(\lambda, \omega)$, we cannot safely apply the UCB selection as in Lemma 4.4. In this case, we set $b_t$ to be the lowest or highest bid, with equal probability, to uniformly explore both sides of the treatment value $\Delta v_t$.

Let $\Phi_{\text{exp}} \subseteq [T]$ denote the times when $b_t$ is selected by the uniform exploration in Line 13 of Algorithm 3. To align with this exploration, we define the modified IPW:

$$\widetilde{e}_t(b) := \begin{cases} 2\mathbb{1}[b \geq m_t]v_{t,1} - 2\mathbb{1}[b < m_t]v_{t,0}, & \text{if } t \in \Phi_{\text{exp}}; \\ \frac{\mathbb{1}[b \geq m_t]v_{t,1}}{\widehat{G}_t(b)} - \frac{\mathbb{1}[b < m_t]v_{t,0}}{1 - \widehat{G}_t(b)}, & \text{otherwise.} \end{cases} \qquad (11)$$

And the corresponding variance proxy:

$$\sigma_t(b) := \begin{cases} 4, & \text{if } t \in \Phi_{\text{exp}}; \\ \frac{1}{\widehat{G}_t(b)(1 - \widehat{G}_t(b))}, & \text{otherwise.} \end{cases} \qquad (12)$$

When $t \in \Phi_{\text{exp}}$, we take advantage of the uniform exploration and use a constant-variance IPW; otherwise we stick to the previous estimator in (5) and (6). Fortunately, since we only require a constant-level precision $C(\lambda, \omega)$, the total exploration times is small:

**Lemma 4.6.** *Suppose the events in Lemma 3.1 and 4.3 hold. Then with probability $1 - T^{-3}$,*

$$|\Phi_{\text{exp}}| = O(d \log^5 T).$$

## 4.6. Regret Analysis

Finally, we piece everything together to arrive at the regret guarantee in Theorem 2.5. Without loss of generality, we will assume that the high probability events in Lemma 3.1, 4.3, and 4.6 hold almost surely. The next result shows that, if the bid $b_t$ is selected at the $\ell$-th level, then the instantaneous suboptimality is $O(2^{-\ell})$. Let $\widehat{b}_t^* = \arg\max_{b \in \mathcal{B}} \overline{r}_t(b)$ denote the hindsight optimal bid in the discretization.

**Lemma 4.7.** *At time $t \in [T]$ and level $\ell \in [L]$, we have $\widehat{b}_t^* \in B_\ell$ and every bid $b \in B_\ell$ satisfies*

$$\overline{r}_t(\widehat{b}_t^*) - \overline{r}_t(b) \le 8 \cdot 2^{-\ell}.$$

By Assumption 2.2, it is straightforward to show that

$$\overline{r}_t(b_t^*) - \overline{r}_t(\widehat{b}_t^*) \le \overline{r}_t(\Delta v_t) - \overline{r}_t\left(\Delta v_t + \frac{1}{\sqrt{T}}\right) \le \frac{1}{\sqrt{T}}. \tag{13}$$

Let $\Phi_{T+1}^{(\ell)}$ denote the time indices associated with level $\ell$ and $\Phi_{T+1}^{(L+1)} = [T] \setminus [(L+1)T_0] \cup \cup_{\ell=1}^L \Phi_{T+1}^{(\ell)}$ the time indices when the bid $b_t$ is selected in Line 19 of Algorithm 3 (i.e. when the confidence widths are small). Let $q_\ell \in \{0,1\}$ denote the UCB index returned by Algorithm 1.

Thanks to the exploration times in Line 13, the widths in Algorithm 1 satisfy $\max_{b^j \in \mathcal{B}} \frac{1-\lambda}{8} \cdot w_t^{(\ell)}(b^j) \le C(\lambda, \omega)$ for all remaining time $t$ and $b^j$, which satisfies the condition of Lemma 4.4. For each $t \in \Phi_{T+1}^{(L+1)}$ selected at level $\ell$,

$$\overline{r}_t(b_t^*) - \overline{r}_t(b_t) \le \widehat{r}_{t,q_\ell}^{(\ell)}(\widehat{b}_t^*) - \arg\max_{b \in B_\ell} \widehat{r}_{t,q_\ell}^{(\ell)} + \frac{2+1}{\sqrt{T}} \le \frac{3}{\sqrt{T}}$$

by Lemma 4.4. Then we can bound the regret as follows:

$$R(\pi) = \mathbb{E}\left[\sum_{t=1}^T \overline{r}_t(b_t^*) - \overline{r}_t(b_t)\right]$$

$$= (L+1)T_0 + \mathbb{E}\left[(3)\sqrt{T} + \sum_{\ell=1}^L \sum_{t \in \Phi_{T+1}^{(\ell)}} \overline{r}_t(b_t^*) - \overline{r}_t(b_t)\right]$$

$$\overset{(a)}{\le} O(\sqrt{T}) + \mathbb{E}\left[|\Phi_{\exp}| + 8\sum_{\ell=1}^L |\Phi_{T+1}^{(\ell)}| 2^{-\ell}\right]$$

$$\overset{(b)}{\le} O(\sqrt{T} + d\log^5 T) + \mathbb{E}\left[8\sum_{\ell=1}^L \sum_{t \in \Phi_{T+1}^{(\ell)}} w_{t,q_\ell}^{(\ell)}(b_t)\right]$$

where (a) is by Lemma 4.7 and (13), and (b) because $w_{t,q_\ell}^{(\ell)}(b_t) > 2^{-\ell}$ in Line 16 when $t \in \Phi_{T+1}^{(\ell)} \setminus \Phi_{\exp}$, and by Lemma 4.6. The proof is completed by the following potential-based lemma and that $w_{t,q_\ell}^{(\ell)}(b_t) = \min\{w_{t,0}^{(\ell)}(b_t), w_{t,1}^{(\ell)}(b_t)\}$ by Lemma 4.4.

**Lemma 4.8.** *For each level $\ell \in [L]$, it holds*

$$\sum_{t \in \Phi_{T+1}^{(\ell)}} \min\{w_{t,0}^{(\ell)}(b_t), w_{t,1}^{(\ell)}(b_t)\} = O(\sqrt{dT}\log^2 T).$$

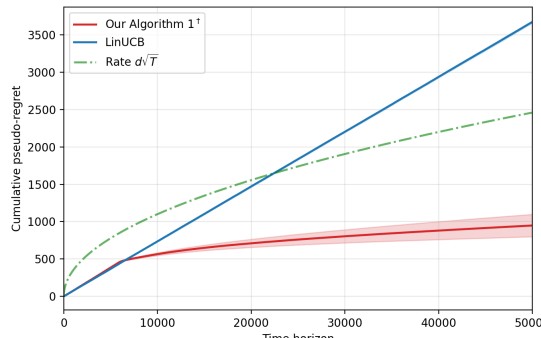

*Figure 2.* **Regret trajectory.** This plots the trajectories of the expected regret of Algorithm 1[†] and the celebrated LinUCB when nonzero baseline outcome $v_{t,0}$ is present. It averages over 10 independent runs, and the shaded region stands for 1 std. LinUCB suffers a linear regret due to consistent overbidding, as expected, by overlooking the baseline outcome and over-estimating the ad value. By contrast, Algorithm 1[†] successfully learns the causal ad value and converges at a desired rate $O(d\sqrt{T})$. During the initial 7000 times, Algorithm 1[†] has not explored sufficiently, so the choice of the width in (9) leads to overbidding and thereby a linearly growing regret. This is an expected behavior, since the information in SPAs is asymmetric, and overbidding is necessary to collect HOB observations when the algorithm is uncertain. More details about the setup is in Appendix A.

## 5. Practical Implementations

So far, we have used a lengthy hierarchical elimination in Algorithm 3 solely to address the dependencies and arrive at the minimax optimal rate $\widetilde{\Theta}(\sqrt{dT})$, which can be too heavy for practice. Therefore, we discard the master routine and integrate necessary components into Algorithm 1 to arrive at a much cleaner variant, Algorithm 1[†]. We remark that this is a common practice in the linear bandit literature (Li et al., 2010; Chu et al., 2011), and the regret bound for the consequent algorithm is typically loosen by a factor of $\sqrt{d}$ (Abbasi-Yadkori et al., 2011). Due to space limit, the details of Algorithm 1[†] and the experiment setup are deferred to Appendix A. Its superiority is demonstrated in Figure 2.

Finally, we remark that the $O(\sqrt{T}\log T)$ initialization can be replaced by estimating CDF $G$ with historical logs. Similarly, the HOB parameters $(\lambda, \omega)$ can be inferred either offline or from the initialization rounds.

## 6. Conclusion

To measure the marginal value of an ad exposure, this work introduces an online causal inference framework to repeated SPAs, widely applied in search engines. By carefully generalizing variance reduction ingredients from the literature and coupling them with the second-price payment, we present a complete picture for the regret characterization across different feedback structures. Our results clarify when and why online causal inference is intrinsically easier in SPAs than in FPAs and how to exploit this advantage optimally.

## Acknowledgements

This work is supported by NSF (CCF-2312205, ECCS-2419564), ONR-13983263 and 2027 New York University Center for Global Economy and Business grant. The authors also gratefully acknowledge Hong Kong Research Grant Council (HKRGC) General Research Fund 16506225, 16308321, and donation grants from Edge Science, Tencent, and Zhuhai Kehui. This research made use of the computing resources of the X-GPU cluster supported by the HKRGC Collaborative Research Fund C6021-19EF.

## Impact Statement

This paper presents work whose goal is to advance the field of Machine Learning. There are many potential societal consequences of our work, none of which we feel must be specifically highlighted here.

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

## A. Numerical Experiment

---

**Algorithm 1[†]: LinUCB.TE.S (treatment value estimation in SPAs)**

---

1   **Input:** Time $T$, discrete bid set $\mathcal{B}$, parameters $(\lambda, \omega)$, tunable learning rate $\eta > 0$.

2   Set $T_0 \leftarrow \lceil \sqrt{T} \log T \rceil$.

3   **for** $t = 1, 2, \ldots, T_0$ **do**

4     Bid $b_t \leftarrow 1$ and observe $m_t$.

5   **end**

6   **for** $t = T_0 + 1, \ldots, T$ **do**

7     Observe context $x_t$.

8     Set variance proxy $\sigma_\tau \leftarrow \sigma_\tau(b_\tau)$ in (12) and selected width $u_\tau \leftarrow u_\tau(b_\tau)$ in (9);

9     $A_t \leftarrow I + \sum_{\tau < t} \sigma_\tau^{-2} x_\tau x_\tau^\top$;

10    $z_t \leftarrow \sum_{\tau < t} \sigma_\tau^{-2} x_\tau \widetilde{e}_\tau$ with IPW $\widetilde{e}_\tau = \widetilde{e}_\tau(b_\tau)$ in (11);

11    Compute estimator $\widehat{\theta}_t \leftarrow A_t^{-1} z_t$.

12    Set $\gamma_t \leftarrow 1 + 14 \log T + 4 \sqrt{\sum_{\tau < t} u_\tau^2}$.

13    Compute CDF estimator $\widehat{G}_t$ as in (4).

14    **for** $b^j \in \mathcal{B}$ **do**

15      Compute the following quantities:

16      $w_{t,0}(b^j) \leftarrow \widehat{G}_t(b^j) \gamma_t \|x_t\|_{A_t^{-1}} + 4 u_t(b^j)$.

17      $w_{t,1}(b^j) \leftarrow (1 - \widehat{G}_t(b^j)) \gamma_t \|x_t\|_{A_t^{-1}} + 4 u_t(b^j)$.

18      $\widehat{r}_{t,0}(b^j) \leftarrow \widehat{G}_t(b^j)(\widehat{\theta}_t^\top x_t - b^j) + \sum_{i \leq j} \widehat{G}_t(b^i)$.

19      $\widehat{r}_{t,1}(b^j) \leftarrow \widehat{r}_{t,0}(b^j) - \widehat{\theta}_t^\top x_t$.

20    **end**

21    Invoke Algorithm 2 with perturbation $c = \frac{\omega}{4}$, margin constant $\epsilon = \frac{1-\lambda}{8}$, CDF $\widehat{G}_t$, and value $\widehat{\theta}_t^\top x_t$ to get interval $I_t$ and index $q \in \{0, 1\}$.

22    **if** $\gamma_t \|x_t\|_{A_t^{-1}} + 4 u_t(b^J) > C(\lambda, \omega)$ **then**

       // Exploration

23      Uniformly randomly select $b_t = 0$ or $1$.

24    **else**

25      Select UCB bid $b_t \leftarrow \arg\max_{b \in \mathcal{B} \cap I_t} \widehat{r}_{t,q}(b) + \eta \cdot w_{t,q}(b)$.

26    **end**

27    Bid $b_t$ and observe either $(v_{t,1}, m_t)$ or $v_{t,0}$.

28   **end**

---

Since most existing real-time bidding datasets do not document the organic clicks (i.e. the baseline outcomes $v_{t,0}$ when lost), we test Algorithm 1[†] empirically on a synthetic environment. In this environment, the baseline $v_{t,0}$ is set to be a nonzero periodic value in time, which aims to reflect the fluctuating market behavior (e.g. from day to night); its pattern is shown in Figure 3. We implement the celebrated `LinUCB` as a benchmark, which only regresses on the winning outcomes as done in current industry (Li et al., 2010). In the presence of nonzero $v_{t,0}$, Figure 2 indicates that `LinUCB` consistently overbids and thereby suffers a linear regret, as expected. By contrast, Algorithm 1[†] successfully identifies the treatment ad value and exhibits a desired convergence rate. This highlights the emergent need for taking causal estimation into account in real-time bidding.

In particular, we consider a simple setup to highlight the overbidding issue of existing methods. At each time $t$, the context $x_t = [1, x_t(2:d)]$ is generated by drawing $x_t(2:d)$ from the $(d-1)$-dimensional isotropic Gaussian $\mathcal{N}_{d-1}(\mathbf{0}, I)$, with dimension $d = 11$, and concatenated with an intercept. Then we normalize by setting $x_t \leftarrow \frac{x_t}{\|x_t\|_2}$. The underlying parameter is $\theta_* = [\theta_*(1), \theta_*(2:d)]$, where we again draw $\theta_*(2:d)$ from $\mathcal{N}_{d-1}(\mathbf{0}, I)$ and set $\theta_*(1) = 0.6$. Then we normalize by $\theta_* \leftarrow$

$\frac{\theta_*}{\|\theta_*\|_2}$. The baseline outcome is periodic in time with a context-dependent fluctuation: $v_{t,0} = \sigma(2 + \sin(f \cdot t) + \cos(\beta^\top x_t))$, where $\sigma$ is the sigmoid function, $f = \frac{\pi}{125}$ gives a length-250 period, and $\beta \sim \mathcal{N}_d(\mathbf{0}, I)$ (and then normalized) is another linear model for the contextual dependence. Its mean $\mathbb{E}_{x_t}[v_{t,0}]$ is illustrated in Figure 3. The periodic pattern is set to reflect the potentially periodic market behaviors; for example, the user conversion is typically high during some fixed 'peak hours' each day, regardless whether the product is displayed in the sponsored slots or high-ranked organic slots. The winning outcome is then set as $v_{t,1} = v_{t,0} + \varepsilon_t$ where $\varepsilon_t \sim \mathrm{Bern}(\theta_*^\top x_t)$ is a Bernoulli random variable. Finally, the HOB is drawn from an i.i.d. Beta distribution $M_t \sim \mathrm{Beta}(5, 7)$.

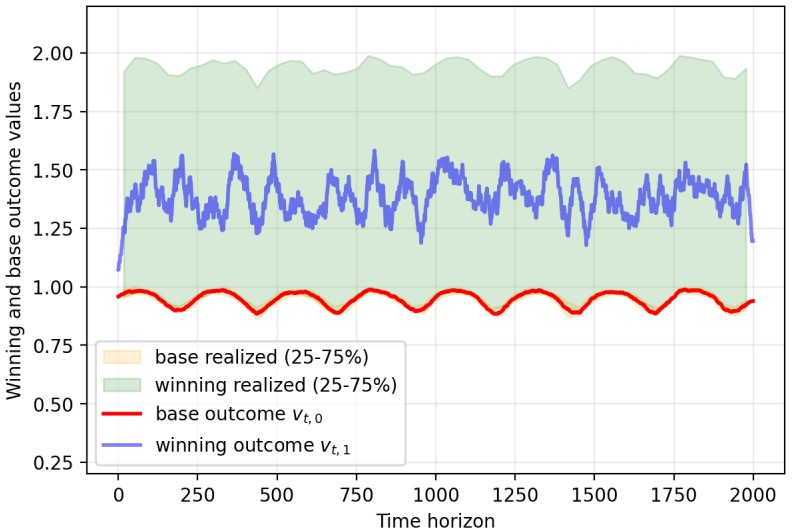

*Figure 3.* **Pattern of periodic outcomes.** The baseline outcome $v_{t,0}$ is defined as a periodic quantity in time. The winning outcome $v_{t,1}$ is then set to $v_{t,0} + \Delta v_t$, where $\Delta v_t$ is a Bernoulli variable with a linear-in-context mean $\theta_*^\top x_t$. This plot shows the realized outcomes $(v_{t,0})_t$ and $(v_{t,1})_t$, smoothed over a window size 35 for visualization. The shaded regions show the 25 to 75 quantile of the realizations in bins of size 35.

## B. Extension to Non-i.i.d. HOBs

As a relaxation of Assumption 2.2, one can consider a contextual HOB that satisfies

$$m_t = f_*(x_t) + \eta_t$$

where $f_* : \mathbb{R}^d \to [0, 1]$ is an underlying function that captures the contextual dependence of the mean, and $\eta_t$ is an i.i.d. noise. For example, a simple yet powerful model is the linear HOB with $f_*(x_t) = \beta_*^\top x_t$ for some unknown $\beta_* \in \mathbb{R}^d$ (Badanidiyuru et al., 2023; Wen et al., 2025a). For this type of contextual HOBs, a natural extension is to apply regressions on the structural part $f_*(x_t)$ and then our analysis on the i.i.d. noise $\eta_t$. Two key challenges remain: The first challenge lies in adapting the bid-space bins in (3) to the *estimated* i.i.d. noise $\widehat{\eta}_t := m_t - \widehat{f}_t(x_t)$, as the learner learns $\widehat{f}_t \to f_*$. When the CDF of $\eta_t$ is Lipschitz, the error from this adaptation may be bounded by $O(\|\widehat{f}_t - f_*\|)$; but in general (e.g. in our case of Assumption 2.2, which includes most continuous/discrete/mixed distributions), it remains challenging (Fuller, 2009; Buonaccorsi, 2010; Wen et al., 2025b). The second challenge is to fully exploit the asymmetric information on $m_t$ in the binary feedback in SPAs, where the regression observations only come from the turns when the bidder wins and pays. Fortunately, our approach already tends to win under large uncertainty to collect information on estimating the i.i.d. part (see the caption in Figure 2 and the UCB quantity in (9)), so this may be less a challenge.

## C. Proof of Lemma 3.1

*Proof.* We will provide a detailed proof for the second (slightly more involved) claim. Then the first claim follows the same argument. Fix any bid index $j \in [[\lceil\sqrt{T}\rceil]]$. Recall that $n_t^j := \sum_{\tau \in \Phi_t} \mathbb{1}[b_\tau \geq b_j]$. Note that by definition in (4),

$$
\frac{1}{\sqrt{T}} \left| \sum_{i \leq j} G(b^i) - \widehat{G}_t(b^i) \right| = \frac{1}{\sqrt{T}} \left| \sum_{i \leq j} \sum_{k \leq i} p^k - \widehat{p}_t^k \right|
$$

$$
= \frac{1}{\sqrt{T}} \left| \sum_{i \leq j} \sum_{k \leq i} \frac{\sum_{\tau \in \Phi_t} \mathbb{1}[b_\tau \geq b^k] (p^k - \mathbb{1}[b^k < m_\tau \leq b^{k+1}])}{\sum_{\tau \in \Phi_t} \mathbb{1}[b_\tau \geq b^k]} \right|
$$

$$
= \frac{1}{\sqrt{T}} \left| \sum_{\tau \in \Phi_t} \sum_{i \leq j} \sum_{k \leq i} \frac{\mathbb{1}[b_\tau \geq b^k]}{n_t^k} (p^k - \mathbb{1}[b^k < m_\tau \leq b^{k+1}]) \right|
$$

$$
= \left| \sum_{\tau \in \Phi_t} \sum_{k \leq j} \frac{j - k}{\sqrt{T}} \frac{\mathbb{1}[b_\tau \geq b^k]}{n_t^k} (p^k - \mathbb{1}[b^k < m_\tau \leq b^{k+1}]) \right|.
$$

To handle this term, denote $X_\tau := \sum_{k \leq j} \frac{j-k}{\sqrt{T}} \frac{\mathbb{1}[b_\tau \geq b^k]}{n_t^k} (p^k - \mathbb{1}[b^k < m_\tau \leq b^{k+1}])$. Clearly $\mathbb{E}[X_\tau] = 0$ as $\mathbb{E}[\mathbb{1}[b^k < m_\tau \leq b^{k+1}]] = p^k$. Additionally, since $n_t^k \geq n_t^j$ for every $k \leq j$ and $\frac{j-k}{\sqrt{T}} \leq 1$, we always have the range of $X_\tau$ be smaller than $\frac{1}{n_t^j}$. Since it is mean-zero, we also have

$$
\mathbb{E}[X_\tau^2] = \text{Var}(X_\tau) \leq \sum_{k \leq j} \frac{(j-k)^2}{T} \text{Var}\left( \frac{\mathbb{1}[b_\tau \geq b^k]}{n_t^k} (p^k - \mathbb{1}[b^k < m_\tau \leq b^{k+1}]) \right)
$$

$$
\leq \sum_{k \leq j} \text{Var}\left( \frac{\mathbb{1}[b_\tau \geq b^k]}{n_t^k} (p^k - \mathbb{1}[b^k < m_\tau \leq b^{k+1}]) \right)
$$

$$
= \sum_{k \leq j} \frac{\mathbb{1}[b_\tau \geq b^k]}{(n_t^k)^2} \cdot \mathbb{E}[(p^k - \mathbb{1}[b^k < m_\tau \leq b^{k+1}])^2]
$$

$$
= \sum_{k \leq j} \frac{\mathbb{1}[b_\tau \geq b^k]}{(n_t^k)^2} p^k (1 - p^k)
$$

$$
\leq \sum_{k \leq j} \frac{\mathbb{1}[b_\tau \geq b^k]}{(n_t^k)^2} p^k.
$$

By the assumption in the lemma statement, the variables $(X_\tau)_{\tau \in \Phi_t}$ are conditionally independent. Consequently,

$$
\sum_{\tau \in \Phi_t} \text{Var}(X_\tau) = \sum_{\tau \in \Phi_t} \mathbb{E}[X_\tau^2] \leq \sum_{\tau \in \Phi_t} \sum_{k \leq j} \frac{\mathbb{1}[b_\tau \geq b^k]}{(n_t^k)^2} p^k = \sum_{k \leq j} \frac{p^k}{n_t^k}.
$$

So by Bernstein's inequality, with probability at least $1 - T^{-4}/2$, we have

$$
\left| \sum_{\tau \in \Phi_t} X_\tau \right| \leq 8 \sqrt{\log(SKT) \sum_{k \leq j} \frac{p^k}{n_t^k}} + \frac{8 \log(SKT)}{n_t^j}. \tag{14}
$$

By taking a union bound over $j \in [[\lceil\sqrt{T}\rceil]]$, with probability at least $1 - T^{-3}/2$, for every $b^j$ we have

$$
\frac{1}{\sqrt{T}} \left| \sum_{i \leq j} G(b^i) - \widehat{G}_t(b^i) \right| \leq 8 \sqrt{\log T \sum_{k \leq j} \frac{p^k}{n_t^k}} + \frac{8 \log T}{n_t^j}. \tag{15}
$$

Since $p^k$ is unknown, we consider the initial estimator $\widehat{p}_0^k$. By (29) and (30) in Appendix C.6 in Han et al. (2025), with probability at least $1 - T^{-4}$, it holds that

$$p^k \leq 2\widehat{p}_0^k + \frac{12 \log T}{\lceil \sqrt{T} \rceil} \tag{16}$$

$$\widehat{p}_0^k \leq 2p^k + \frac{12 \log T}{\lceil \sqrt{T} \rceil}. \tag{17}$$

Combining (15) and (16) gives

$$\frac{1}{\sqrt{T}} \left| \sum_{i \leq j} G(b^i) - \widehat{G}_t(b^i) \right| \leq 8 \sqrt{\log T \sum_{k \leq j} \frac{2}{n_t^k} \left( \widehat{p}_0^k + \frac{12 \log T}{\lceil \sqrt{T} \rceil} \right)} + \frac{8 \log T}{n_t^j}.$$

The same bound applies to the first claim on $\left| G(b^j) - \widehat{G}_t(b^j) \right|$: By definition, for each bid index $j \in [\lceil \sqrt{T} \rceil]$,

$$\left| G(b^j) - \widehat{G}_t(b^j) \right| = \left| \sum_{i \leq j} p^i - \widehat{p}_t^i \right| = \left| \sum_{i \leq j} \frac{\sum_{\tau \in \Phi_t} \mathbb{1}[b_\tau \geq b^i] (p^i - \mathbb{1}[b^i < m_\tau \leq b^{i+1}])}{\sum_{\tau \in \Phi_t} \mathbb{1}[b_\tau \geq b^i]} \right|$$

$$= \left| \sum_{\tau \in \Phi_t} \sum_{i \leq j} \frac{\mathbb{1}[b_\tau \geq b^i]}{n_t^i} (p^i - \mathbb{1}[b^i < m_\tau \leq b^{i+1}]) \right|.$$

Following the same argument, with probability at least $1 - T^{-3}/2$, for every $b^j$ we have

$$\left| G(b^j) - \widehat{G}_t(b^j) \right| \leq 8 \sqrt{\log T \sum_{k \leq j} \frac{p^k}{n_t^k}} + \frac{8 \log T}{n_t^j} \tag{18}$$

which can be combined with (16) as well. Taking the union bound over (15), (18), and (16) completes the proof. $\square$

## D. Proof of Lemma 4.1

*Proof.* First consider the bias of this estimator. By definition in (5), we have

$$\left| \mathbb{E}[\widetilde{e}_t(b)] - \theta_*^\top x_t \right| = \left| \mathbb{E} \left[ \frac{\mathbb{1}[b \geq M_t]}{\widehat{G}_t(b)} v_{t,1} - v_{t,1} - \frac{\mathbb{1}[b < M_t]}{1 - \widehat{G}_t(b)} v_{t,0} + v_{t,0} \right] \right|$$

$$\leq \left| \mathbb{E} \left[ \frac{\mathbb{1}[b \geq M_t]}{\widehat{G}_t(b)} - 1 \right] \right| + \left| \mathbb{E} \left[ \frac{\mathbb{1}[b < M_t]}{1 - \widehat{G}_t(b)} - 1 \right] \right|$$

$$= \frac{\left| G(b) - \widehat{G}_t(b) \right|}{\widehat{G}_t(b)} + \frac{\left| G(b) - \widehat{G}_t(b) \right|}{1 - \widehat{G}_t(b)}$$

$$\overset{(a)}{\leq} u_t(b) \left( \frac{1}{\widehat{G}_t(b)} + \frac{1}{1 - \widehat{G}_t(b)} \right)$$

$$\overset{(b)}{\leq} 2 u_t(b) \sigma_t(b)$$

where (a) applies $\left| \widehat{G}_t(b) - G(b) \right| \leq u_t(b)$ by assumption, and (b) applies $\frac{1}{\widehat{G}_t(b)} + \frac{1}{1 - \widehat{G}_t(b)} \leq \frac{2}{\min\{\widehat{G}_t(b), 1 - \widehat{G}_t(b)\}} \leq \frac{2}{\widehat{G}_t(b)(1 - \widehat{G}_t(b))}$.

Next, the bound on variance is as follows:

$$\text{Var}(\widetilde{e}_t(b)) \leq \mathbb{E}[\widetilde{e}_t(b)^2] \overset{(c)}{\leq} \frac{2\mathbb{E}[\mathbb{1}[b \geq M_t]]}{\widehat{G}_t(b)^2} + \frac{2\mathbb{E}[\mathbb{1}[b < M_t]]}{(1 - \widehat{G}_t(b))^2}$$

$$\leq \frac{2}{\widehat{G}_t(b)^2} + \frac{2}{(1 - \widehat{G}_t(b))^2}$$

$$\leq 4\sigma_t(b)^2$$

where (c) follows from the AM-GM inequality $(a + b)^2 \leq 2a^2 + 2b^2$. □

## E. Proof of Lemma 4.3

*Proof.* Let the bias of the IPW estimator in (5) at time $\tau$ be $\zeta_\tau := \mathbb{E}[\widetilde{e}_\tau(b_\tau)] - \theta_*^\top x_\tau$. During the remaining of this proof, we denote $D_t = [\sigma_\tau^{-1} x_\tau]_{\tau \in \Phi_t} \in \mathbb{R}^{d \times |\Phi_t|}$ as the weighted contexts, $V_t = [\sigma_\tau^{-1} \widetilde{e}_\tau(b_\tau)]_{\tau \in \Phi_t} \in \mathbb{R}^{|\Phi_t| \times 1}$ the weighted estimators, and $Z_t = [\sigma_\tau^{-1} \zeta_\tau]_{\tau \in \Phi_t} \in \mathbb{R}^{|\Phi_t| \times 1}$ the weighted biases, where $\sigma_\tau^{-1} = \sigma_\tau(b_\tau)^{-1} = \widehat{G}_t(b_\tau)\left(1 - \widehat{G}_t(b_\tau)\right)$ as defined in (6). Recall that in Algorithm 1, we have $A_t = I + D_t D_t^\top$. With $\widehat{\theta}_t$ defined in Algorithm 1, we can decompose the value estimation error by

$$\widehat{\theta}_t^\top x_t - \theta_*^\top x_t = x_t^\top A_t^{-1} D_t V_t - x_t^\top A_t^{-1}(I + D_t D_t^\top)\theta_*$$

$$= x_t^\top A_t^{-1} D_t(V_t - Z_t - D_t^\top \theta_*) + x_t^\top A_t^{-1} D_t Z_t - x_t^\top A_t^{-1} \theta_*.$$

Since $\|\theta_*\|_2 \leq 1$ and $|x_t^\top A_t^{-1} \theta_*| \leq \|x_t^\top A_t^{-1}\|_2$, this gives

$$\left|\widehat{\theta}_t^\top x_t - \theta_*^\top x_t\right| \leq \left|x_t^\top A_t^{-1} D_t(V_t - Z_t - D_t^\top \theta_*)\right| + \left|x_t^\top A_t^{-1} D_t Z_t\right| + \|x_t^\top A_t^{-1}\|_2. \tag{19}$$

Since $A_t = I + D_t D_t^\top \succeq I$, the last term is upper bounded by

$$\|x_t^\top A_t^{-1}\|_2 = \sqrt{x_t^\top A_t^{-1} I A_t^{-1} x_t} \leq \sqrt{x_t^\top A_t^{-1} x_t} = \|x_t\|_{A_t^{-1}}.$$

Next we address the first term in (19). Note that

$$x_t^\top A_t^{-1} D_t(V_t - Z_t - D_t^\top \theta_*) = \sum_{\tau \in \Phi_t} x_t^\top A_t^{-1} \sigma_\tau^{-2} x_\tau \widetilde{\varepsilon}_\tau$$

where the noise $\widetilde{\varepsilon}_\tau = \widetilde{e}_\tau(b_\tau) - \theta_*^\top x_\tau - \zeta_\tau$ satisfies the followings:

- $(\widetilde{\varepsilon}_\tau)_{\tau \in \Phi_t}$ are conditionally independent given $(x_\tau, b_\tau, M_\tau)_{\tau \in \Phi_t}$ by the lemma statement;

- $\mathbb{E}[\widetilde{\varepsilon}_\tau] = 0$;

- $|\widetilde{\varepsilon}_\tau| \leq |\widetilde{e}_\tau(b_\tau)| \leq 2\max\left\{\widehat{G}_\tau(b_\tau)^{-1}, \left(1 - \widehat{G}_\tau(b_\tau)\right)^{-1}\right\} \leq 2\sigma_\tau$;

- $\text{Var}(\widetilde{\varepsilon}_\tau) = \text{Var}(\widetilde{e}_\tau(b_\tau)) \leq 4\sigma_\tau^2$ by Lemma 4.1.

Then for every $\tau \in \Phi_t$, we have

- $\text{Var}\left(x_t^\top A_t^{-1} \sigma_\tau^{-2} x_\tau \widetilde{\varepsilon}_\tau\right) \leq \left|x_t^\top A_t^{-1} \sigma_\tau^{-1} x_\tau\right|^2 \sigma_\tau^{-2} \text{Var}(\widetilde{\varepsilon}_\tau) \leq 4\left|x_t^\top A_t^{-1} \sigma_\tau^{-1} x_\tau\right|^2$;

- $\left|x_t^\top A_t^{-1} \sigma_\tau^{-2} x_\tau \widetilde{\varepsilon}_\tau\right| \leq 2\left|x_t^\top A_t^{-1} x_\tau\right| \leq 2\|x_t^\top A_t^{-1}\|_2 \leq 2\|x_t\|_{A_t^{-1}}$.

By Bernstein's inequality (Lemma K.1), with probability at least $1 - T^{-3}$,

$$
\left| \sum_{\tau \in \Phi_t} x_t^\top A_t^{-1} \sigma_\tau^{-2} x_\tau \widetilde{\varepsilon}_\tau \right| \leq \sqrt{8 \log(2T^3) \sum_{\tau \in \Phi_t} \left| x_t^\top A_t^{-1} \sigma_\tau^{-1} x_\tau \right|^2} + \frac{4}{3} \log(2T^3) \|x_t\|_{A_t^{-1}}
$$

$$
= \sqrt{8 \log(2T^3) \|x_t^\top A_t^{-1} D_t\|_2^2} + \frac{4}{3} \log(2T^3) \|x_t\|_{A_t^{-1}}
$$

$$
\overset{(a)}{\leq} \sqrt{8 \log(2T^3) \|x_t\|_{A_t^{-1}}^2} + \frac{4}{3} \log(2T^3) \|x_t\|_{A_t^{-1}}
$$

$$
\leq 14 \log T \|x_t\|_{A_t^{-1}}
$$

where (a) follows from $\|x_t^\top A_t^{-1} D_t\|_2^2 = x_t^\top A_t^{-1} D_t D_t^\top A_t^{-1} x_t \leq x_t A_t^{-1} x_t = \|x_t\|_{A_t^{-1}}$ as $A_t = I + D_t D_t^\top \succeq D_t D_t^\top$.

Finally, we bound the middle term in (19). We can write

$$
\left| x_t^\top A_t^{-1} D_t Z_t \right| \overset{(b)}{\leq} \|x_t A_t^{-1}\|_{A_t} \|D_t Z_t\|_{A_t^{-1}} = \|x_t\|_{A_t^{-1}} \|D_t Z_t\|_{A_t^{-1}} = \|x_t\|_{A_t^{-1}} \sqrt{Z_t^\top D_t^\top A_t^{-1} D_t Z_t}
$$

$$
\overset{(c)}{\leq} \|x_t\|_{A_t^{-1}} \|Z_t\|_2
$$

where (b) applies Cauchy-Schwartz inequality and (c) applies Woodbury matrix identity with $D_t^\top A_t D_t = D_t^\top (I + D_t D_t^\top)^{-1} D_t \preceq I$. Applying Lemma 4.1 on the following term completes the proof:

$$
\|Z_t\|_2 = \sqrt{\sum_{\tau \in \Phi_t} \sigma_\tau^{-2} \zeta_\tau^2} \leq 4 \sqrt{\sum_{\tau \in \Phi_t} u_\tau^2}.
$$

$\square$

## F. Proof of Lemma 4.4

To be explicit, the constant in Lemma 4.4 is (recall we used a perturbation level $c = \frac{\omega}{4}$ in Algorithm 2)

$$
C(\lambda, \omega) = \frac{c}{2} \cdot \frac{1 - \lambda}{8} = \frac{\omega(1 - \lambda)}{64}.
$$

The proof of this lemma consists of three parts. First, we show that the confidence widths $w_{t,0}$ and $w_{t,1}$ are valid via Lemma F.1. Then we show in Lemma F.2 that, over the interval found by Algorithm 2, the estimated CDF value is bounded away from 1, and the optimal bid lies in the chosen interval. Finally, we show that the chosen index $q \in \{0, 1\}$ gives the smaller width in Lemma F.3.

**Lemma F.1.** *Suppose the event in Lemma 3.1 holds. Also suppose $|G(b^j) - \widehat{G}_t(b^j)| \leq u_t(b^j)$ and*

$$
\frac{1}{\sqrt{T}} \left| \sum_{i \leq j} G(b^i) - \sum_{i \leq j} \widehat{G}_t(b^i) \right| \leq u_t(b^j)
$$

*for each $b^j \in \mathcal{B}$ for the given width function $u_t$ in Algorithm 1. Then it holds that*

$$
|\overline{r}_{t,0}(b) - \widehat{r}_{t,0}(b)| \leq \frac{1 - \lambda}{8} \cdot w_{t,0}(b)
$$

*and*

$$
|\overline{r}_{t,1}(b) - \widehat{r}_{t,1}(b)| \leq \frac{1 - \lambda}{8} \cdot w_{t,1}(b)
$$

*for every discretized bid $b \in \mathcal{B}$.*

*Proof.* We will present the proof for the first formulation $\overline{r}_{t,0}$, as the proof for the other formulation follows verbatim. Fix any $b^j \in \mathcal{B}$. By definition, we have

$$
|\overline{r}_{t,0}(b^j) - \widehat{r}_{t,0}(b^j)| = \left| G(b^j)\theta_*^\top x_t - G(b^j)b^j + \int_0^{b^j} G(m)\mathrm{d}m - \widehat{G}_t(b^j)\widehat{\theta}_t^\top x_t + \widehat{G}(b^j)b^j - \frac{1}{\sqrt{T}}\sum_{i \leq j} \widehat{G}_t(b^i) \right|
$$

$$
\leq \left| G(b^j)(\theta_*^\top x_t - b^j) - \widehat{G}_t(b^j)(\theta_*^\top x_t - b^j) \right| + \left| \widehat{G}_t(b^j)\theta_*^\top x_t - \widehat{G}_t(b^j)\widehat{\theta}_t^\top x_t \right|
$$

$$
+ \left| \int_0^{b^j} G(m)\mathrm{d}m - \frac{1}{\sqrt{T}}\sum_{i \leq j} G(b^i) \right| + \left| \frac{1}{\sqrt{T}}\sum_{i \leq j} G(b^i) - \frac{1}{\sqrt{T}}\sum_{i \leq j} \widehat{G}_t(b^i) \right|.
$$

To proceed, we address each term separately. First,

$$
\left| G(b^j)(\theta_*^\top x_t - b^j) - \widehat{G}_t(b^j)(\theta_*^\top x_t - b^j) \right| \leq |G(b^j) - \widehat{G}_t(b^j)| \leq u_t(b^j) \tag{20}
$$

since $|\theta_*^\top x_t - b^j| \leq 1$. Similarly,

$$
\left| \widehat{G}_t(b^j)\theta_*^\top x_t - \widehat{G}_t(b^j)\widehat{\theta}_t^\top x_t \right| = \widehat{G}_t(b^j)\left| \theta_*^\top x_t - \widehat{\theta}_t^\top x_t \right| \leq \widehat{G}_t(b^j)\gamma \|x_t\|_{A_t^{-1}} \tag{21}
$$

by Lemma 4.3. The third term is bounded as

$$
\left| \int_0^{b^j} G(m)\mathrm{d}m - \frac{1}{\sqrt{T}}\sum_{i \leq j} G(b^i) \right| = \frac{1}{\sqrt{T}}\sum_{i \leq j} G(b^i) - \int_0^{b^j} G(m)\mathrm{d}m
$$

$$
= \frac{1}{\sqrt{T}}\sum_{i \leq j} G(b^i) - \sum_{i \leq j} \int_{b^{i-1}}^{b^i} G(m)\mathrm{d}m
$$

$$
\overset{(a)}{\leq} \sum_{i \leq j} \frac{G(b^i) - G(b^{i-1})}{\sqrt{T}}
$$

$$
= \frac{G(b^j)}{\sqrt{T}} \leq \frac{1}{\sqrt{T}} \tag{22}
$$

where (a) because $b^i - b^{i-1} = \frac{1}{\sqrt{T}}$. The last term is bounded by $u_t(b^j)$ by the statement assumption. Combining this with (20–22) and the definition of $w_{t,0}$ completes the proof. $\qquad\square$

**Lemma F.2.** *Let $c = \frac{\omega}{4}$, $\epsilon = \frac{1-\lambda}{8}$, and $[b_{\mathrm{L}}, b_{\mathrm{R}}]$ be the interval found in Algorithm 2. It holds that*

*(1) $\arg\max_{b \in \mathcal{B}} \overline{r}_t(b) \in [b_{\mathrm{L}}, b_{\mathrm{R}}]$;*

*(2) $\widehat{G}_t(b_{\mathrm{R}}) - \widehat{G}_t(b_{\mathrm{L}}) \leq 1 - \epsilon$, for any $\epsilon \in (0, 1)$, when*

$$
\left| \widehat{v} - \theta_*^\top x_t \right| + \arg\max_{b^j \in \mathcal{B}} \left| G(b^j) - \widehat{G}_t(b^j) \right| + \left| \int_0^{b^j} G(m)\mathrm{d}m - \frac{1}{\sqrt{T}}\sum_{i \leq j} \widehat{G}_t(b^i) \right| \leq \min\left\{ \frac{c \cdot \epsilon}{2}, \frac{1 - 4\epsilon - \lambda}{2} \right\}.
$$

*Proof.* Recall the definitions in Algorithm 2 that the perturbed bid optimizers are $b_+ = \arg\max_{b^j \in \mathcal{B}} \widehat{G}_t(b)(\widehat{v} + c) - \frac{1}{\sqrt{T}}\sum_{i \leq j} \widehat{G}_t(b^i)$ and $b_- = \arg\max_{b^j \in \mathcal{B}} \widehat{G}_t(b)(\widehat{v} - c) - \frac{1}{\sqrt{T}}\sum_{i \leq j} \widehat{G}_t(b^i)$. The set is $S = \{b \in \mathcal{B} : \widehat{G}_t(b_-) - c \leq \widehat{G}_t(b) \leq \widehat{G}_t(b_+) + c\}$ and the final endpoints are $b_{\mathrm{L}} = \min S$, and $b_{\mathrm{R}} = \max S$. Write $b^{j*} = \arg\max_{b \in \mathcal{B}} \overline{r}_t(b)$ and

$v_t = \theta_*^\top x_t$ for simplicity. We have

$$G_t(b^{j*})(v_t - b^{j*}) + \int_0^{b^{j*}} G(m)\mathrm{d}m \leq \widehat{G}_t(b^{j*})(v_t - b^{j*}) + \left|G(b^{j*})(v_t - b^{j*}) - \widehat{G}_t(b^{j*})(v_t - b^{j*}) + \int_0^{b^{j*}} G(m)\mathrm{d}m\right|$$

$$\leq \widehat{G}_t(b^{j*})(v_t - b^{j*}) + \frac{1}{\sqrt{T}}\sum_{i \leq j_*}\widehat{G}_t(b^i) + \arg\max_{b^j \in \mathcal{B}}\left|G(b^j) - \widehat{G}_t(b^j)\right| + \left|\int_0^{b^j} G(m)\mathrm{d}m - \frac{1}{\sqrt{T}}\sum_{i \leq j}\widehat{G}_t(b^i)\right|$$

$$\leq \widehat{G}_t(b^{j*})(\widehat{v}_t - b^{j*}) + \frac{1}{\sqrt{T}}\sum_{i \leq j_*}\widehat{G}_t(b^i) + |\widehat{v}_t - v_t| + \arg\max_{b^j \in \mathcal{B}}\left|G(b^j) - \widehat{G}_t(b^j)\right| + \left|\int_0^{b^j} G(m)\mathrm{d}m - \frac{1}{\sqrt{T}}\sum_{i \leq j}\widehat{G}_t(b^i)\right|$$

$$\leq \widehat{G}_t(b^{j*})(\widehat{v}_t - c - b^{j*}) + \frac{1}{\sqrt{T}}\sum_{i \leq j_*}\widehat{G}_t(b^i) + |\widehat{v}_t - v_t| + \arg\max_{b^j \in \mathcal{B}}\left|G(b^j) - \widehat{G}_t(b^j)\right| + \left|\int_0^{b^j} G(m)\mathrm{d}m - \frac{1}{\sqrt{T}}\sum_{i \leq j}\widehat{G}_t(b^i)\right| + c\widehat{G}_t(b^{j*})$$

$$\leq \widehat{G}_t(b_-)(\widehat{v}_t - c - b_-) + \frac{1}{\sqrt{T}}\sum_{i \leq j(b_-)}\widehat{G}_t(b^i) + |\widehat{v}_t - v_t| + \arg\max_{b^j \in \mathcal{B}}\left|G(b^j) - \widehat{G}_t(b^j)\right| + \left|\int_0^{b^j} G(m)\mathrm{d}m - \frac{1}{\sqrt{T}}\sum_{i \leq j}\widehat{G}_t(b^i)\right| + c\widehat{G}_t(b^{j*})$$

$$\leq G(b_-)(\widehat{v}_t - b_-) + \int_0^{b_-} G(m)\mathrm{d}m + |\widehat{v}_t - v_t| + \arg\max_{b^j \in \mathcal{B}} 2\left|G(b^j) - \widehat{G}_t(b^j)\right| + 2\left|\int_0^{b^j} G(m)\mathrm{d}m - \frac{1}{\sqrt{T}}\sum_{i \leq j}\widehat{G}_t(b^i)\right| + c(\widehat{G}_t(b^{j*}) - \widehat{G}_t(b_-))$$

$$\leq G(b_-)(v_t - b_-) + \int_0^{b_-} G(m)\mathrm{d}m + 2|\widehat{v}_t - v_t| + \arg\max_{b^j \in \mathcal{B}} 2\left|G(b^j) - \widehat{G}_t(b^j)\right| + 2\left|\int_0^{b^j} G(m)\mathrm{d}m - \frac{1}{\sqrt{T}}\sum_{i \leq j}\widehat{G}_t(b^i)\right| + c(\widehat{G}_t(b^{j*}) - \widehat{G}_t(b_-))$$

where we denote the index of the bid $b_- \in \mathcal{B}$ as $b_- = b^{j(b_-)}$. Since $G_t(b_-)(v_t - b_-) + \int_0^{b_-} G(m)\mathrm{d}m \leq G_t(b^{j*})(v_t - b^{j*}) + \int_0^{b^{j*}} G(m)\mathrm{d}m$, this implies

$$\widehat{G}_t(b^{j*}) \geq \widehat{G}_t(b_-) - \frac{2}{c}\left(|\widehat{v} - v_t| + \arg\max_{b^j \in \mathcal{B}}\left|G(b^j) - \widehat{G}_t(b^j)\right| + \left|\int_0^{b^j} G(m)\mathrm{d}m - \frac{1}{\sqrt{T}}\sum_{i \leq j}\widehat{G}_t(b^i)\right|\right)$$

$$\geq \widehat{G}_t(b_-) - \epsilon$$

where the second inequality comes from the lemma assumption. The other direction holds under a symmetric argument, giving $\widehat{G}_t(b^{j*}) \leq \widehat{G}_t(b_+) + \epsilon$. Therefore, $b^{j*} \in [b_\mathrm{L}, b_\mathrm{R}]$ by definition.

For the second claim, since $\mathcal{B} \subseteq [0,1]$ is a $\frac{1}{\sqrt{T}}$−discretization, Lemma K.5 implies $\widehat{G}_t(b_+) - \widehat{G}_t(b_-) \leq 1 - 4\epsilon$. Finally, we have

$$\widehat{G}_t(b_\mathrm{R}) - \widehat{G}_t(b_\mathrm{L}) \leq 1 - 4\epsilon + 2\epsilon \leq 1 - \epsilon$$

as desired. $\qquad\square$

**Lemma F.3.** *Suppose the conditions in Lemma 4.4 hold. For the index $q \in \{0,1\}$ returned by Algorithm 2 with perturbation $c = \frac{\omega}{4}$, it holds that $w_{t,q}(b) = \min\{w_{t,0}(b), w_{t,1}(b)\}$ for every bid $b \in [b_\mathrm{L}, b_\mathrm{R}]$, and $\arg\max_{b \in \mathcal{B}} \overline{r}_t(b) \in [b_\mathrm{L}, b_\mathrm{R}]$.*

*Proof.* Recall that $q = \mathbb{1}[\widehat{G}_t(b_\mathrm{L}) \geq \epsilon]$ with $\epsilon = \frac{1-\lambda}{8}$. By Lemma F.2, it holds that $\arg\max_{b \in \mathcal{B}} \overline{r}_t(b) \in [b_\mathrm{L}, b_\mathrm{R}]$ and $\widehat{G}_t(b_\mathrm{R}) - \widehat{G}_t(b_\mathrm{L}) \leq 1 - 2\epsilon$. Consider a case study. Suppose $q = 0$ and $\widehat{G}_t(b_\mathrm{L}) < \epsilon$, which implies $\widehat{G}_t(b_\mathrm{R}) < 1 - \epsilon$. Then for every $b \in [b_\mathrm{L}, b_\mathrm{R}]$,

$$\widehat{G}_t(b) < 1 - \epsilon \implies \frac{1 - \widehat{G}_t(b)}{\epsilon} > 1 > \widehat{G}_t(b).$$

Plugging in this to the definitions of $w_{t,0}$ and $w_{t,1}$ in Algorithm 1 shows

$$\epsilon \cdot w_{t,0}(b) \leq \min\{w_{t,0}(b), w_{t,1}(b)\}.$$

A similar argument for $q = 1$ completes the proof. $\qquad\square$

# G. Proof of Lemma 4.6

*Proof.* Fix any level $\ell \in [L]$ for now and suppress the superscript $(\ell)$ for the ease of notation. Recall that $\max\{w_{t,0}(b), w_{t,1}(b)\} \leq \frac{8}{1-\lambda}\left(\gamma\|x_t\|_{A_t^{-1}} + 4u_t(b) + \frac{2}{\sqrt{T}}\right)$. It suffices to bound the number of times (for this level) when $\gamma\|x_t\|_{A_t^{-1}} > \frac{4}{1-\lambda}C(\lambda, \omega)$ or $4u_t(b) > \frac{4}{1-\lambda}C(\lambda, \omega)$. For simplicity, let constant $c_0 := \frac{4}{1-\lambda}C(\lambda, \omega)$ and $\Phi_{\exp}(\ell) \subseteq \Phi_{\exp}$ be the exploration times attributed to this level.

First, we consider the linear term $\gamma\|x_t\|_{A_t^{-1}}$ and let

$$\Phi^1_{\exp}(\ell) := \sum_{t \in \Phi_{\exp}(\ell)} \mathbb{1}\left[\gamma\|x_t\|_{A_t^{-1}} > c_0\right].$$

Since $\gamma = 1 + 14\log T + 4\sqrt{\sum_{\tau \in \Phi_t} u_\tau^2}$ is an increasing coefficient in time, consider the final coefficient $\gamma_T$ at the last time. By (25), $\gamma \leq \gamma_T = O(\log^{\frac{3}{2}} T)$. We have

$$c_0|\Phi^1_{\exp}(\ell)| < \sum_{t \in \Phi_{\exp}(\ell)} \gamma_t\|x_t\|_{A_t^{-1}} \leq \gamma_T \sum_{t \in \Phi_{\exp}(\ell)} \|x_t\|_{A_t^{-1}} \stackrel{(a)}{=} 4\gamma_T \sum_{t \in \Phi_{\exp}(\ell)} \|\sigma_t^{-1}x_t\|_{A_t^{-1}} \stackrel{(b)}{=} O\left(\sqrt{d|\Phi^1_{\exp}(\ell)|}\log^2 T\right),$$

where (a) follows from the definition in (12), and (b) from Lemma K.3. This gives $|\Phi^1_{\exp}(\ell)| = O(d\log^4 T)$.

Next, since $u_t(b^i) \leq u_t(b^j)$ for bids $i < j$, we consider

$$\Phi^2_{\exp,t}(\ell) := \sum_{\tau \in \Phi_{\exp}(\ell):\tau < t} \mathbb{1}\left[u_\tau(b^J) > c_0\right]$$

where $b^J = 1$ is the largest bid. Note that $u_\tau(b^J) > c_0$ implies $n_\tau^J < c_0'\log T$ for some other constant $c_0' = O(c_0^2)$, by definition in (9). Since the exploration was uniform over $b^1 = 0$ and $b^J = 1$, at every time $t$, the standard Chernoff bound gives us

$$n_t^J \geq \frac{|\Phi^2_{\exp,t}(\ell)|}{2} - 2\sqrt{|\Phi^2_{\exp,t}(\ell)|\log T}$$

with probability at least $1 - T^{-4}$. By a union bound, this event holds for all $t \in [T]$ with probability $1 - T^{-3}$. Suppose $|\Phi^2_{\exp,t}(\ell)| \geq \max\{64\log T, 4c_0'\log T\}$ at some $t$, which implies $n_t^J \geq \frac{|\Phi^2_{\exp,t}(\ell)|}{4}$ from the above high-probability bound. Then we also have $n_t^J \geq c_0'\log T$, so $u_t(b^J) \leq c_0$ and subsequently $|\Phi^2_{\exp,t}(\ell)| = \cdots = |\Phi^2_{\exp,T+1}(\ell)|$ for all future time (since $n_t^J$ is increasing in time). Finally, as $|\Phi^2_{\exp,t}(\ell)|$ increments by 1, it holds that $|\Phi^2_{\exp}(\ell)| \leq \max\{64\log T, 4c_0'\log T\} = O(\log T)$.

The desired bound then follows from

$$|\Phi_{\exp}| = \sum_{\ell=1}^{L} |\Phi_{\exp}(\ell)| \leq \sum_{\ell=1}^{L} \left(|\Phi^1_{\exp}(\ell)| + |\Phi^2_{\exp}(\ell)|\right) = O(L \cdot d\log^4 T) = O(d\log^5 T).$$

$\square$

# H. Proof of Lemma 4.7

*Proof.* This result is proved via an inductive argument over the levels $\ell \in [L]$. For the sake of clarity, the inductive hypothesis is: $\widehat{b}_t^* \in B_\ell$ and $\overline{r}_t(\widehat{b}_t^*) - \widehat{r}_t(b) \leq 8 \cdot 2^{-\ell}$. For $\ell = 1$, the claim trivially holds by the boundedness of $\overline{r}_t$. Now suppose the claim holds up to the level $\ell - 1$.

For clarity, let $q_\ell \in \{0, 1\}$ denote the UCB index returned by Algorithm 1 at level $\ell$. When Algorithm 3 reached the elimination step in Line 22 at level $\ell - 1$, it holds that

$$w_{t,q_{\ell-1}}^{(\ell-1)}(b) \leq 2^{-(\ell-1)} \text{ for every } b \in B_{\ell-1}.$$

Since $\widehat{b}_t^* \in B_{\ell-1}$, by Lemma F.1, it holds that

$$\widehat{r}_{t,q_{\ell-1}}^{(\ell-1)}(\widehat{b}_t^*) \geq \overline{r}_{t,q_{\ell-1}}(\widehat{b}_t^*) - 2^{-(\ell-1)} \geq \overline{r}_{t,q_{\ell-1}}(b) - 2^{-(\ell-1)} \geq \widehat{r}_{t,q_{\ell-1}}^{(\ell-1)}(b) - 2 \cdot 2^{-(\ell-1)}.$$

Thanks to the exploration condition in Line 13 of Algorithm 3, the confidence width condition in Lemma F.2 (2) holds. Then together with Lemma F.2 that $\widehat{b}_t^* \in \mathcal{I}_t^{(\ell-1)}$, this implies that $\widehat{b}_t^*$ is not eliminated, i.e. $\widehat{b}_t^* \in B_\ell$. To see the second claim in the hypothesis, note that for every $b \in B_\ell$,

$$\begin{aligned}
\overline{r}_t(\widehat{b}_t^*) - \overline{r}_t(b) &= \overline{r}_{t,q_{\ell-1}}(\widehat{b}_t^*) - \overline{r}_{t,q_{\ell-1}}(b) \\
&\leq \widehat{r}_{t,q_{\ell-1}}(\widehat{b}_t^*) - \widehat{r}_{t,q_{\ell-1}}(b) + w_{t,q_{\ell-1}}^{(\ell-1)}(\widehat{b}_t^*) + w_{t,q_{\ell-1}}^{(\ell-1)}(b) \\
&\leq \widehat{r}_{t,q_{\ell-1}}(\widehat{b}_t^*) - \widehat{r}_{t,q_{\ell-1}}(b) + 2 \cdot 2^{-(\ell-1)} \\
&\overset{(a)}{\leq} 4 \cdot 2^{-(\ell-1)} = 8 \cdot 2^{-\ell}
\end{aligned}$$

where (a) follows from the elimination criterion for $B_\ell$ that

$$\max_{b' \in B_{\ell-1}} \widehat{r}_{t,q_{\ell-1}}(b') - 2 \cdot 2^{-(\ell-1)} \leq \widehat{r}_{t,q_{\ell-1}}(b) \leq \max_{b' \in B_{\ell-1}} \widehat{r}_{t,q_{\ell-1}}(b')$$

for every $b \in B_\ell$ and, in particular, $\widehat{b}_t^*$. $\qquad\square$

## I. Proof of Lemma 4.8

*Proof.* Let $b_t = b^{j_t} \in \mathcal{B}$. Fix any level $\ell \in [L]$ and we suppress the notation on $\ell$ for the notations (e.g. $\Phi_t^{(\ell)}$, $u_t^{(\ell)}$, $\gamma$, $A_t$) throughout the remaining proof. Let $\gamma_t$ denote the parameter $\gamma = 1 + 14\log T + 4\sqrt{\sum_{\tau \in \Phi_t} u_\tau^2}$ in Algorithm 1. Recall from the definitions that

$$\sum_{\tau \in \Phi_t} \min\{w_{t,0}(b_t), w_{t,1}(b_t)\} = \frac{8}{1-\lambda} \sum_{\tau \in \Phi_t} \left( \min\{\widehat{G}_t(b_t), 1 - \widehat{G}_t(b_t)\} \gamma_t \|x_t\|_{A_t^{-1}} + 4u_t(b_t) + \frac{2}{\sqrt{T}} \right). \tag{23}$$

To proceed, we consider the sum over each term respectively. First, recall that $A_t = I + \sum_{\tau \in \Phi_t} \sigma_\tau^{-2} x_\tau x_\tau^\top$ where $\sigma_\tau^{-1} = \widehat{G}_\tau(b_\tau)(1 - \widehat{G}_\tau(b_\tau))$. Then

$$\begin{aligned}
\sum_{t \in \Phi_{T+1}} \min\{\widehat{G}_t(b_t), 1 - \widehat{G}_t(b_t)\} \gamma_t \|x_t\|_{A_t^{-1}} &\leq \sum_{t \in \Phi_{T+1}} 2\widehat{G}_t(b_t)(1 - \widehat{G}_t(b_t)) \gamma_T \|x_t\|_{A_t^{-1}} \\
&= 2 \sum_{t \in \Phi_{T+1}} \gamma_T \|\sigma_t^{-1} x_t\|_{A_t^{-1}} \\
&\overset{(a)}{\leq} \gamma_T \sqrt{8d|\Phi_{T+1}| \log\left(1 + \frac{|\Phi_{T+1} - 1|}{d}\right)} \\
&\overset{(b)}{\leq} \sqrt{8dT} \log^2 T \tag{24}
\end{aligned}$$

where (a) follows Lemma K.3. To see (b), recall that the initialization guarantees the number of observations $n_t^j \geq \sqrt{T} \log T$ for all $t \in \Phi_{T+1}$. By definition in (9), it holds that

$$\gamma_T = 1 + 14 \log T + 4 \sqrt{\sum_{t \in \Phi_{T+1}} u_t^2}$$

$$\overset{(c)}{=} O\left( \log T + \sqrt{\sum_{t \in \Phi_{T+1}} \sum_{k \leq j_t} \frac{\log T}{n_t^k} \left( \widehat{p}_0^k + \frac{\log T}{\sqrt{T}} \right) + \sum_{t \in \Phi_{T+1}} \frac{\log^2 T}{(n_t^{j_t})^2}} \right)$$

$$\overset{(d)}{\leq} O\left( \log T + \sqrt{\sum_{t \in \Phi_{T+1}} \sum_{k \leq j_t} \frac{\log T}{n_t^k} \left( p^k + \frac{\log T}{\sqrt{T}} \right) + \sum_{t \in \Phi_{T+1}} \frac{\log^2 T}{T \log^2 T}} \right)$$

$$\leq O\left( \log T + \sqrt{1 + \sum_{t \in \Phi_{T+1}} \sum_{k \leq j_t} \frac{\log T}{n_t^k} \left( p^k + \frac{\log T}{\sqrt{T}} \right)} \right)$$

$$\overset{(e)}{\leq} O\left( \log T + \sqrt{1 + \log^3 T} \right) = O(\log^{\frac{3}{2}} T) \tag{25}$$

where (c) applies the elementary inequality $(a + b)^2 \leq 2a^2 + 2b^2$, (d) applies (17), and (e) invokes Lemma K.4 with $v_j = p^j + \log T / \sqrt{T}$. The second term is bounded similarly: by Cauchy-Schwartz inequality,

$$\sum_{t \in \Phi_{T+1}} u_t(b_t) \leq \sqrt{T} \sqrt{\sum_{t \in \Phi_{T+1}} u_t(b_t)^2} \leq \gamma_T \sqrt{T} = O(\sqrt{T} \log^{\frac{3}{2}} T). \tag{26}$$

Plugging (24) and (26) into (23) completes the proof. $\qquad\square$

## J. Proof of Lower Bound

In this section, we prove the lower bound in Theorem 2.6. At a high level, we divide the horizon equally into $d$ equal subhorizons and embed an independent lower bound instance to each of them. First, we have an existing non-contextual lower bound (also see (Weed et al., 2016)):

**Theorem J.1.** *Let $v_{t,0} \equiv 0$ and i.i.d. HOB that follows a known distribution:*

$$m_t \sim \frac{1}{2} \mathrm{Unif}[0, 1] + \frac{1}{2} \delta_{\frac{1}{4} + \Delta}$$

*where $\delta_m$ denotes the point mass at $m_t = m$ and $\Delta = \frac{1}{4\sqrt{T}}$. Consider a special family of instances where $v_{t,1} \sim \mathrm{Bern}(\mu)$ for an unknown mean $\mu$. Then*

$$\inf_{\pi} \sup_{\mu \in [0,1]} \mathbb{E}\left[ \sum_{t=1}^{T} \max_{b^* \in [0,1]} \overline{r}_t(b^*) - \overline{r}_t(b_t) \right] = \Omega(\sqrt{T}).$$

*Proof.* This proof will use Le Cam's two-point lower bound. Let $\mu_1 = \frac{1}{4}$ and $\mu_2 = \frac{1}{4} + 2\Delta$. Let $\overline{r}_t^{(i)}(b) = G(b)(\mu_i - b) + \int_0^b G(m) \mathrm{d}m$ be the expected payoff for each setting $i = 1, 2$. Clearly, the maximizing bids are

$$\arg\max_{b \in [0,1]} \overline{r}_t^{(1)}(b) = \mu_1 \text{ and } \arg\max_{b \in [0,1]} \overline{r}_t^{(2)}(b) = \mu_2$$

respectively (Vickrey, 1961). Now we show that no bid can perform well under *both* settings. Without loss of generality, we

may focus on bids $b_t \in [\mu_1, \mu_2] = \left[\frac{1}{4}, \frac{1}{4} + 2\Delta\right]$. For any bid $\frac{1}{4} \leq b_t < \frac{1}{4} + \Delta$, we have

$$
\begin{aligned}
\overline{r}_t^{(1)}(\mu_1) - \overline{r}_t^{(1)}(b_t) + \overline{r}_t^{(2)}(\mu_2) - \overline{r}_t^{(2)}(b_t) &\geq \overline{r}_t^{(2)}(\mu_2) - \overline{r}_t^{(2)}(b_t) \\
&= \int_0^{\mu_2} G(m)\mathrm{d}m - G(b_t)(\mu_2 - b_t) - \int_0^{b_t} G(m)\mathrm{d}m \\
&= \int_{b_t}^{\mu_2} (G(m) - G(b_t))\mathrm{d}m \\
&\geq \Delta \cdot \frac{1}{2} = \frac{\Delta}{2}
\end{aligned}
\tag{27}
$$

where the last line follows from the construction of $G$ and that $b_t < \frac{1}{4} + \Delta \leq \mu_2 - \Delta$. Similarly, if $\frac{1}{4} + \Delta \leq b_t \leq \frac{1}{4} + 2\Delta$, then

$$
\begin{aligned}
\overline{r}_t^{(1)}(\mu_1) - \overline{r}_t^{(1)}(b_t) + \overline{r}_t^{(2)}(\mu_2) - \overline{r}_t^{(2)}(b_t) &\geq \overline{r}_t^{(1)}(\mu_1) - \overline{r}_t^{(1)}(b_t) \\
&= \int_0^{\mu_1} G(m)\mathrm{d}m - G(b_t)(\mu_1 - b_t) - \int_0^{b_t} G(m)\mathrm{d}m \\
&= \int_{\mu_1}^{b_t} (G(b_t) - G(m))\mathrm{d}m \\
&\geq \Delta \cdot \frac{1}{2} = \frac{\Delta}{2}.
\end{aligned}
\tag{28}
$$

Now fix any policy $\pi$. For each environment defined by $\mu_i$ with $i = 1, 2$, the interaction with $\pi$ gives rise to the distribution $\mathbb{P}_i^{\otimes T}$ over the horizon $T$. We denote the environment-specific regret as

$$
R_i(\pi) = \mathbb{E}_{\mathbb{P}_i^{\otimes T}}\left[\sum_{t=1}^T \max_{b^* \in [0,1]} \overline{r}_t^{(i)}(b^*) - \overline{r}_t^{(i)}(b_t)\right]
$$

where the bids $b_t$ are chosen by policy $\pi$. Standard Le Cam analysis leads to:

$$
\begin{aligned}
R_1(\pi) + R_2(\pi) &= \sum_{t=1}^T \mathbb{P}_1^{\otimes t}\left(\max_{b^* \in [0,1]} \mathbb{E}[\overline{r}_t^{(1)}(b^*) - \overline{r}_t^{(1)}(b_t)]\right) + \mathbb{P}_2^{\otimes t}\left(\max_{b^* \in [0,1]} \mathbb{E}[\overline{r}_t^{(2)}(b^*) - \overline{r}_t^{(2)}(b_t)]\right) \\
&\overset{(a)}{\geq} \frac{\Delta}{2} \sum_{t=1}^T \int \min\{\mathrm{d}\mathbb{P}_1^{\otimes t}, \mathrm{d}\mathbb{P}_2^{\otimes t}\} \\
&\overset{(b)}{=} \frac{\Delta}{2} \sum_{t=1}^T \left(1 - \|\mathbb{P}_1^{\otimes t} - \mathbb{P}_2^{\otimes t}\|_{\mathrm{TV}}\right) \\
&\overset{(c)}{\geq} \frac{\Delta T}{2}\left(1 - \|\mathbb{P}_1^{\otimes T} - \mathbb{P}_2^{\otimes T}\|_{\mathrm{TV}}\right)
\end{aligned}
$$

where (a) is by (27) and (28), (b) uses $\int \min\{\mathrm{d}P, \mathrm{d}Q\} = 1 - \|P - Q\|_{\mathrm{TV}}$, and (c) follows from the data processing inequality for the total variation distance. By the chain rule of KL divergence and an elementary inequality for KL divergence between Bernoulli distributions, it holds that

$$
D_{\mathrm{KL}}\left(\mathrm{Bern}(p)^{\otimes T} \| \mathrm{Bern}(q)^{\otimes T}\right) = T \cdot D_{\mathrm{KL}}(\mathrm{Bern}(p)\|\mathrm{Bern}(q)) \leq T\frac{(p-q)^2}{q(1-q)}.
$$

Then by Lemma K.2, we arrive at

$$
R_1(\pi) + R_2(\pi) \geq \frac{\Delta T}{4} \exp\left(-cT\Delta^2\right).
$$

for some absolute constant $c > 0$. Finally, plugging in the choice of $\Delta = \frac{1}{4\sqrt{T}}$ leads to

$$
\max_{i=1,2} R_i(\pi) \geq \frac{R_1(\pi) + R_2(\pi)}{2} = \Omega(\sqrt{T}),
$$

which completes the proof. $\square$

Next, we prove the contextual lower bound in Theorem 2.6.

*Proof of Theorem 2.6.* The proof proceeds by embedding the noncontextual lower bound instance in Theorem J.1 in the contextual case in Theorem 2.6. Again, consider the special case where $v_{t,0} \equiv 0$.

Suppose first $d \geq 2$, and let

$$\theta_* = \left(\frac{1}{2}, \text{Unif}\left(\left\{0, 4\Delta\right\}^{d-1}\right)\right),$$

with $\Delta = \frac{1}{4}\sqrt{\frac{d-1}{T}}$. Since $T \geq d^2$, it holds that $\|\theta_*\|_2 \leq 1$. We divide the time horizon $T$ into $d-1$ sub-horizons $T_n$ for $n = 1, \ldots, d-1$ with equal length $\frac{T}{d-1}$. The context during the sub-horizon $T_n$ is chosen to be

$$x_t = \left(\frac{1}{2}, 0, \ldots, 0, \frac{1}{2}, 0, \ldots, 0\right)$$

where the second $\frac{1}{2}$ appears in the $(n+1)$-th entry. The HOB distribution for $m_t$ is again the i.i.d. distribution in Theorem J.1. Note that by construction, each sub-horizon becomes an independent learning sub-problem. Therefore, we can decompose the regret $R(\pi)$ into the sum of regrets from $d-1$ independent sub-problems, each of time duration $\frac{T}{d-1}$. By Theorem J.1, we have

$$\inf_{\pi} R(\pi) = (d-1) \cdot \Omega\left(\sqrt{\frac{T}{d-1}}\right) = \Omega(\sqrt{dT}).$$

Finally, consider the case $d = 1$. We simply let

$$\theta_* \sim \text{Unif}\left(\left\{\frac{1}{4}, \frac{1}{4} + 2\Delta\right\}\right)$$

with $\Delta = \frac{1}{4\sqrt{T}}$ and $x_t \equiv 1$. Then applying Theorem J.1 again gives the desired regret lower bound $\Omega(\sqrt{T})$. $\qquad\square$

## K. Auxiliary Lemmata

**Lemma K.1** (Bernstein's inequality in (Boucheron et al., 2003))**.** *Consider independent random variables $X_1, \ldots, X_n \in [a, b]$. We have*

$$\mathbb{P}\left(\left|\sum_{i=1}^{n} X_i - \sum_{i=1}^{n} \mathbb{E}[X_i]\right| \geq \varepsilon\right) \leq 2\exp\left(-\frac{\varepsilon^2}{2(\sigma^2 + \varepsilon(b-a)/3)}\right)$$

*for any $\varepsilon > 0$, where $\sigma^2 = \sum_{i=1}^{n} \text{Var}(X_i)$.*

*In particular, it implies the following confidence bound: for any $\delta \in (0, 1)$, with probability at least $1 - \delta$, we have*

$$\frac{1}{n}\left|\sum_{i=1}^{n} X_i - \sum_{i=1}^{n} \mathbb{E}[X_i]\right| \leq \sqrt{\frac{2\sigma^2/n\log(2/\delta)}{n}} + \frac{2(b-a)\log(2/\delta)}{3n}.$$

**Lemma K.2** (Bretagnolle–Huber inequality (Bretagnolle & Huber, 1978))**.** *Let $P, Q$ be two probability measures on the same probability space. Then*

$$1 - \|P - Q\|_{\text{TV}} \geq \frac{1}{2}\exp(-D_{\text{KL}}(P\|Q))$$

*where $\|\cdot\|_{\text{TV}}$ denotes the total variation distance, and $D_{\text{KL}}$ denotes the KL divergence.*

**Lemma K.3** (Elliptical potential lemma (Abbasi-Yadkori et al., 2011; Wen et al., 2025a))**.** *For any given vectors $\{z_\tau\}_{\tau=1}^{t-1}$ in $\mathbb{R}^d$ with $\|z_\tau\|_2 \leq 1$, let the Gram matrix be $A_s = I + \sum_{\tau<s} z_\tau z_\tau^\top$ and for every $1 \leq s \leq t$. It holds that*

$$\sum_{\tau<t} \|z_\tau\|_{A_\tau^{-1}}^2 \leq 2d\log\left(1 + \frac{t-1}{d}\right).$$

*In particular, by Cauchy-Schwartz inequality,*

$$\sum_{\tau < t} \|z_\tau\|_{A_\tau^{-1}} \le \sqrt{2d(t-1)\log\left(1 + \frac{t-1}{d}\right)}.$$

**Lemma K.4** (Lemma 16 in Han et al. (2025)). *Let $n_t^j$ be defined as in* (10) *for any fixed level in Algorithm 3 and $(v_1, \dots, v_J)$ be a sequence of any nonnegative numbers such that $\sum_{j=1}^{J} v_j = s$. Then it holds that*

$$\sum_{t \in \Phi_{T+1}} \sum_{j \le j_t} \frac{v_j}{n_t^j} \le s(1 + \log T).$$

### K.1. Auction-related Auxiliary Lemmata

**Lemma K.5** (Bounded Optimizer Gap under Value Perturbation). *Let $G$ be a $(\omega, \lambda)$-locally-bounded CDF on $[0,1]$ with $\omega, \lambda \in (0,1)$ (c.f. Definition 2.4), and $\widehat{G}$ be another CDF with $\sup_{b \in \mathcal{B}} |G(b) - \widehat{G}(b)| \le \frac{1-\epsilon-\lambda}{2}$ for some $\epsilon > 0$. Let $\mathcal{B} \subseteq [0,1]$ be a $\frac{1}{\sqrt{T}}$-discretization with $\sqrt{T} > \frac{4}{\omega}$. Denote $\widehat{b}_*(v) = \arg\max_{b \in \mathcal{B}} \widehat{G}(b)(v-b) + \int_0^b \widehat{G}(m) \mathrm{d}m$ with tie broken by taking the bid closest to the value $v$. For any $v_1 \le v_2$, if $v_2 - v_1 \le \frac{\omega}{2}$, then*

$$|\widehat{G}(\widehat{b}_*(v_2)) - \widehat{G}(\widehat{b}_*(v_1))| \le 1 - \epsilon.$$

*Proof.* Since we are in an SPA, it is known that $\widehat{b}_*(v_1) = v_1$ and $\widehat{b}_*(v_2) = v_2$ when the bid space is continuous (Vickrey, 1961). Here $\mathcal{B}$ is a $\frac{1}{\sqrt{T}}$-discretization, so we have $|\widehat{b}_*(v_1) - v_1| \le \frac{1}{\sqrt{T}}$ and $|\widehat{b}_*(v_2) - v_2| \le \frac{1}{\sqrt{T}}$, and also $\widehat{b}_*(v_1) \le \widehat{b}_*(v_2)$. For the sake of simplicity, write $b_1 = \widehat{b}_*(v_1)$ and $b_2 = \widehat{b}_*(v_2)$. By monotonicity of $\widehat{G}$, we have $\widehat{G}(b_1) \le \widehat{G}(b_2)$. For the sake of contradiction, assume $\widehat{G}(b_2) - \widehat{G}(b_1) > 1 - \lambda$. Since $|b_1 - b_2| \le |v_1 - v_2| + \frac{2}{\sqrt{T}} < \omega$ and $G$ is $(\omega, \lambda)$-locally-bounded, we have

$$1 - \epsilon < \widehat{G}(b_2) - \widehat{G}(b_1) \le (1 - \epsilon - \lambda) + G(b_2) - G(b_1) \le 1 - \epsilon,$$

which gives a contradiction. This completes the proof.

$\square$

