# OpenReview forum: "The (Marginal) Value of a Search Ad: An Online Causal Framework for Repeated Second-price Auctions"
_ICML.cc/2026/Conference — ICML 2026 regular_

### Official Review · Reviewer_To5G · 2026-03-09

**Soundness:** 4
**Presentation:** 4
**Significance:** 3
**Originality:** 3
**Overall Recommendation:** 5
**Confidence:** 3

**Summary:**

This paper studies bidding in repeated second-price auctions for search advertising under a causal notion of ad value. Instead of treating the value of an ad opportunity as the outcome obtained after winning, the paper focuses on the incremental effect of winning the ad auction relative to losing it. Under binary feedback, the paper develops online learning algorithms that exploit the payment rule in second-price auctions: when the bidder wins, the payment reveals the highest other bid and provides additional information for learning. In particular, observations from higher bids can be reused for lower bids, so smaller bids effectively have more samples; this is crucial for improving the regret dependence from $T^{2/3}$ in the corresponding first-price setting to $\sqrt{T}$ here. The method also uses the “better of two upper confidence bounds (UCBs)” idea to handle unreliable treatment-effect estimation and control regret. The paper proves a $\tilde O(\sqrt{dT})$ regret guarantee for its algorithm, and also shows that this rate is essentially optimal.

**Compliance With Llm Reviewing Policy:**

Affirmed.

**Final Justification:**

The rebuttal resolved my questions. I decided to recommend Accept.

**Key Questions For Authors:**

Q1. Can the authors position the paper more explicitly relative to recent work on uplift-based bidding? The paper is clearly motivated by incremental value estimation, but it would be helpful to clarify how the present formulation relates to more recent uplift-bidding in advertising practice and in the literature. And how should readers interpret the connection between this theory and practical uplift-based bid policies?

Q2. Can the authors compare more directly against value-based bidding and uplift-based bidding rules, at least conceptually or on synthetic data? Since the practical motivation is that outcome-based bidding may overestimate true value when there is organic substitution, it would strengthen the paper to better illustrate when the proposed causal-value framework leads to materially different bidding behavior or utility.

Q3. How compatible is the framework with context-dependent or non-i.i.d. auction environments? The current analysis assumes i.i.d. HOBs, while the treatment effect is modeled conditionally through context. Could the authors comment on whether the main ideas might extend to settings where the HOB distribution is context-dependent?

**Limitations:**

yes

**Strengths And Weaknesses:**

Strengths
S1. Significance & Practicality: The paper addresses a highly meaningful problem in search advertising. The causal view of ad value is well-motivated, as it acknowledges that sponsored slot losses do not equate to zero outcomes due to the presence of organic results. This bridges the gap between standard auction-learning and incremental/uplift-style bidding.
S2. Technical Soundness: The technical development is rigorous. I particularly appreciate the effective use of the SPA payment rule: the insight that higher bids reveal information that can be reused for lower bids is a clever approach to improving regret.
S3. Theoretical Depth: The main regret result under binary feedback is strong, and the matching lower bound makes the characterization essentially tight. A particularly appealing part is the use of the “better of two upper confidence bounds (UCBs)” idea: when the estimated propensity/CDF leads to potentially large value-estimation error, the algorithm can still rely on the smaller of two confidence formulations so that the effective regret contribution remains controlled.
S4. Presentation: The algorithmic presentation is logical, moving clearly from HOB estimation to treatment-effect estimation and finally to the bidding algorithm.

Weaknesses
W1. Connection to uplift bidding: The paper is clearly motivated by incremental value estimation, but its relationship to uplift bidding in advertising is not discussed as explicitly as it could be.
W2. A strong and meaningful extension of prior work: A main contribution of the paper is the claim that under binary feedback, the SPA setting admits improved regret relative to the FPA setting because the second-price payment rule reveals additional information upon winning. But I think the full-information vs. binary-feedback distinction could be described more precisely. In particular, the paper groups the setting under “binary feedback,” although in SPAs the winner also observes the HOB via the payment rule like right-censor rather than both-censor. More broadly, compared with Wen et al. (2025a), this paper provides a reasonable extension to the SPA-specific information structure and relaxes the required HOB error condition, but several key ingredients remain similar, including the minimax lower-bound style, the weighted least-squares treatment of heteroskedasticity, and the “better of two upper confidence bounds (UCBs)” mechanism. I therefore view the paper as a strong and meaningful extension rather than a fully new framework.
W3. The practical side is underdeveloped: Despite the strong advertising motivation, the paper includes no empirical evaluation, even on synthetic data. This makes it difficult to judge how large the practical benefit is relative to simpler outcome-based bidding baselines or uplift-style heuristics. At the same time, the theoretical model relies on assumptions such as i.i.d. HOBs and a linear treatment-effect model, and it does not address practical constraints such as budget pacing, return-on-spend targets, delayed outcomes, or richer auction dynamics. These limitations do not invalidate the theory contribution, but they do narrow the paper’s practical significance and should be discussed more clearly.

---

> ### Author Rebuttal · Authors · 2026-03-30
>
> We appreciate the reviewer’s positive remark on the contributions and soundness of this study and questions about the current manuscript.
>
> **Comparison to FPA:** We agree with the reviewer that this work can be viewed as a strong and meaningful extension to the SPAs instead of a fully new one, since it shares some core ideas with previous works. On the other hand, we would like to highlight that this problem setup particularly fits search ads, since the sponsored slots appear side-by-side with the organic results, and search currently largely adopts SPAs instead of FPAs. The extension is also highly nontrivial in theory as we summarized in the contributions.
>
> Additionally, as the reviewer correctly notes, SPAs observe extra information from the payment even under the binary feedback, so it is not really “binary”. This is the key distinction that makes SPAs fundamentally easier than FPAs (in terms of learning), as we try to highlight throughout this work.
>
> **Connection to uplift:** We thank the reviewer for bringing this missing related literature to our attention. Indeed, there are a few existing studies on uplift or incrementality bidding that are related to our work, and we will include them in the revision. To be concrete, a number of previous works have proposed to model the incremental gain as the value of a display in real-time bidding (RTB), indicating the increasing attention to this problem. Nonetheless, most of them focus on the empirical perspective [1,2,6,7] due to the theoretical difficulties. Others rely on either massive offline data from randomized logging policy [3], a random bidding control experiment [4,5], or an often unrealistic overlap condition (that is, the probability of both winning and losing is bounded away from 0 and 1 by a constant at all times) [8] to estimate the uplift/incrementality value, and then apply standard RTB algorithms. Among these works, only [8] provides a regret analysis. We are the first to develop provably optimal algorithms without such restrictive assumptions in SPAs.
>
> **Experiments:** Due to space limit, we respectfully refer the reviewer to our response to Reviewer ctVz for more details on the synthetic experiments. In a word:
> - When the bidder receives **nonzero** baseline outcome, LinUCB suffers a constant mis-specification error from **overbidding**, for not considering the causal ad value, while our algorithm converge quickly as expected.
>
> | Time | Our Algorithm 1 mean | Our Algorithm 1 std | LinUCB mean | LinUCB std |
> |---:|---:|---:|---:|---:|
> | 1000 | 74.73 | 2.60 | 73.56 | 1.46 |
> | 5000 | 384.99 | 6.36 | 367.46 | 3.30 |
> | 15000 | 649.03 | 36.51 | 1102.66 | 9.57 |
> | 30000 | 803.89 | 88.09 | 2203.02 | 9.90 |
> | 50000 | 948.04 | 149.96 | 3672.68 | 14.50 |
>
> **Extension to non-iid HOBs:** This is an excellent question! Indeed, the current analysis and HOB estimation crucially rely on the iid assumption, as we treat the CDF value at bid $b^j$ invariant over time. A natural extension is to consider the linear HOB: $m_t = \beta_*^\top x_t + \eta_t$ where the noise $\eta_t$ is iid over time and the HOB mean admits a linear (or other context-dependent) structure. Then we can estimate $\beta_*$ on the fly and adopt the current HOB estimation to the iid noise $\eta_t$. In general, if HOB admits such a structure, one can apply the current approach to the iid part and apply regression to the structured part.
>
> We thank the reviewer again for the valuable comments, and we appreciate any follow-up questions or potential reconsideration of the score.
>
> ---
> References:
>
> [1] Gordon, Brett R., et al. "A comparison of approaches to advertising measurement: Evidence from big field experiments at Facebook." Marketing Science 38.2 (2019): 193-225.
>
> [2] Lewis, Randall, and Jeffrey Wong. "Incrementality bidding and attribution." arXiv preprint arXiv:2208.12809 (2022).
>
> [3] Bompaire, Martin, Alexandre Gilotte, and Benjamin Heymann. "Causal models for real time bidding with repeated user interactions." Proceedings of the 27th ACM SIGKDD Conference on Knowledge Discovery & Data Mining. 2021.
>
> [4] Johnson, Garrett A., Randall A. Lewis, and Elmar I. Nubbemeyer. "Ghost ads: Improving the economics of measuring online ad effectiveness." Journal of Marketing Research 54.6 (2017): 867-884.
>
> [5] Xu, Jian, et al. "Lift-based bidding in ad selection." Proceedings of the aaai conference on artificial intelligence. Vol. 30. No. 1. 2016.
>
> [6] Gordon, Brett R., Robert Moakler, and Florian Zettelmeyer. "Predictive incrementality by experimentation (pie) for ad measurement." arXiv preprint arXiv:2304.06828 (2023).
>
> [7] Waisman, Caio, Harikesh S. Nair, and Carlos Carrion. "Online causal inference for advertising in real-time bidding auctions." Marketing Science 44.1 (2025): 176-195.
>
> [8] Badanidiyuru Varadaraja, Ashwinkumar, et al. "Incrementality bidding via reinforcement learning under mixed and delayed rewards." Advances in Neural Information Processing Systems 35 (2022): 2142-2153.

---

> > ### Author Rebuttal · Reviewer_To5G · 2026-04-03
> >
> > The authors’ rebuttal has effectively addressed several of my primary concerns. In particular, the added reference and discussion is helpful to clarify the positioning relative to existing uplift/incrementality bidding literature. The proposed analytical pathway for extending the framework to context-dependent/non-i.i.d. HOB environments significantly enhances the work’s practical relevance. Furthermore, the newly added synthetic experiments align closely with the theoretical predictions and convincingly demonstrate the algorithm’s advantage over outcome-based baselines.
> > Given the paper’s strong technical soundness and clear presentation, I now view this paper as a well-motivated and theoretically solid extension of prior work to causal ad valuation in second-price auctions. I am therefore updating my Overall Recommendation to 5 (Accept).
> > For the camera-ready version, I recommend formally incorporating the non-i.i.d. discussion and the synthetic experimental results to ensure the completeness and impact of the work.

---

> > > ### Author Response · Authors · 2026-04-03
> > >
> > > Thank you for the interesting questions and recognition. We will be sure to include these changes and the experiments in the revision.

---

### Official Review · Reviewer_ctVz · 2026-03-12

**Soundness:** 3
**Presentation:** 3
**Significance:** 2
**Originality:** 3
**Overall Recommendation:** 4
**Confidence:** 3

**Summary:**

This paper proposes an online causal inference framework for binary auctions in digital advertising. This framework estimates the marginal value of ad exposure (i.e., the difference between winning and losing the auction) and uses this value to make bids. The core contribution lies in designing a learning algorithm that cleverly utilizes information revealed by the binary auction payoff rules, achieving an upper bound of regret $\widetilde{O}(\sqrt{dT})$ under binary feedback. This demonstrates that binary auctions are inherently easier to learn than single-price auctions when estimating treatment effects and making bids.

**Compliance With Llm Reviewing Policy:**

Affirmed.

**Final Justification:**

Having reviewed the original manuscript and the authors’ rebuttal, I maintain my original Weak accept recommendation, as the authors have addressed the concerns raised in my initial review, resolved my prior questions regarding the work’s soundness and clarity.

**Key Questions For Authors:**

1. All conclusions are theoretical analyses. While the theory is highly complete, for a paper involving practical applications (computational advertising), including simulation experiments or validation on public datasets would greatly enhance its persuasiveness and demonstrate the algorithm's performance under actual parameters and its robustness to deviations from assumptions.

2. Although the paper solves an important problem in the field, its impact currently remains primarily theoretical. The proposed algorithmic framework (especially the hierarchical main routine and complex HOB estimation) may have high computational and engineering complexity in practice. The paper does not discuss computational efficiency, storage overhead, or online inference time, which are factors that must be considered in practical deployments. Therefore, its "translation" to industrial practice may require further engineering simplifications or approximations.

**Limitations:**

yes

**Strengths And Weaknesses:**

Strengths:

1. The paper is technically rigorous. Its theoretical framework is clearly constructed. The proofs of the main theorems are logically complete and the steps are detailed.

2. The paper is very clearly written and logically structured.

3. Clear importance and practicality: The paper solves an important problem in the field of automated mechanism design that has been identified but not well addressed by the classic AMA framework: designing high-yield and property-guaranteed auctions when bidder valuations are relevant.

Weaknesses:

1. All conclusions are theoretical analyses. While the theory is highly complete, for a paper involving practical applications (computational advertising), including simulation experiments or validation on public datasets would greatly enhance its persuasiveness and demonstrate the algorithm's performance under actual parameters and its robustness to deviations from assumptions.

2. Although the paper solves an important problem in the field, its impact currently remains primarily theoretical. The proposed algorithmic framework (especially the hierarchical main routine and complex HOB estimation) may have high computational and engineering complexity in practice. The paper does not discuss computational efficiency, storage overhead, or online inference time, which are factors that must be considered in practical deployments. Therefore, its "translation" to industrial practice may require further engineering simplifications or approximations.

---

> ### Author Rebuttal · Authors · 2026-03-30
>
> We appreciate the reviewer’s positive remark on the contributions of this study and questions about the current manuscript.
>
> **Practical Implementation:** As the reviewer highlighted, the hierarchical elimination scheme in Algorithm 3 is purely a theoretical device for handling possible dependencies (c.f. Section 4.4). In the literature, a common practice is to implement only the base routine (in our case, Algorithm 1) and leave the master routine only for theory. The HOB estimators, on the other hand, are somewhat necessary to approximate the second-price payment which is itself an integral. Once the discretization grid of the bid space is fixed, the implementation only requires tracking the counts $1[m_t\le b^j]$ over time and is relatively straightforward. The computation efficiency scales in the same manner as the celebrated LinUCB, since we will only use `np.max()`, `np.argmax()`, and `np.cumsum()` for HOB. Nonetheless, it does take some engineering efforts and theoretical familiarity to implement UCB selection in Algorithm 2, which is the price we pay due to the intrinsic complexity of this causal problem.
>
> It is worth noting that, without the master routine, a regret of order $\widetilde{O}(d\sqrt{T})$ is typically achievable, which is loose by a fact $\sqrt{d}$. The theory there is to control the parameter error $\|\hat\theta_t - \theta_*\|$ over *all* directions in $\mathbb{R}^d$, as opposed to over the single direction $x_t$ in Lemma 4.3. We refer to Abbasi-Yadkori et al. 2011 as a reference. This work uses the (more complicated) master routine simply to derive a complete theoretical picture as in Table 1.
>
> **Experiments:** We appreciate the reviewer’s point on the practicality, and we have run experiments of our Algorithm 1 on synthetic data (because most existing bidding datasets do not collect the losing outcome from organic results for the bidders, as far as we know). We also implemented the celebrated LinUCB algorithm as a benchmark. The setup is as follows:
> The baseline outcome $v_{t,0}$ can be nonzero and is a periodic function in time, which resembles the periodic user behavior (e.g. from day to night). Context $x_t$ enters the periodic value with a small coefficient and nonlinearly.
> The treatment value $v_{t,1}-v_{t,0}$ is a Bernoulli variable with a mean $\theta_*^\top x_t$.
>
> The observations are as follows:
> - When the bidder receives **nonzero** baseline outcome from losing (i.e. from organic results), LinUCB suffers a constant mis-specification error from **overbidding**, because it does not take into account the causal nature of this marginal ad value. Our algorithm is able to identify the treatment value $\theta_*$ and converge quickly as expected, even though the baseline outcome is far from linear in context.
> - When there is **no** baseline outcome, our algorithm suffers a larger regret in the intial stage compared to LinUCB. After an initial stage, the convergence rate of our algorithm can match that of LinUCB. We note that when there is offline data to estimate the HOB, it is possible to bypass the initialization in this work.
>
> These synthetic numerical results will be included in the revision. Since revised manuscript cannot be updated during rebuttal, we include a table of the numerical results here when there is **nonzero** baseline outcome: Mean and std are computed over 10 independent runs.
>
> | Time | Our Algorithm 1 mean | Our Algorithm 1 std | LinUCB mean | LinUCB std |
> |---:|---:|---:|---:|---:|
> | 1000 | 74.73 | 2.60 | 73.56 | 1.46 |
> | 5000 | 384.99 | 6.36 | 367.46 | 3.30 |
> | 15000 | 649.03 | 36.51 | 1102.66 | 9.57 |
> | 30000 | 803.89 | 88.09 | 2203.02 | 9.90 |
> | 50000 | 948.04 | 149.96 | 3672.68 | 14.50 |
>
> From this table, LinUCB suffers a linear regret due to consistent overbidding, as expected, by overlooking the baseline outcome and over-estimating the ad value. By contrast, our algorithm successfully learns the causal ad value and converges at a desired rate $O(d\sqrt{T})$. Nonetheless, note that during the initial $5000$ times, our algorithm is slightly worse than LinUCB. The reason is that it has not explored sufficiently, so the choice of the width in Eq. (9) leads to overbidding and thereby a linearly growing regret. This is intentional, since the information in SPAs is asymmetric, and overbidding is necessary to collect HOB observations when the algorithm is uncertain.
>
> To improve the practical impact, we will also include a clean algorithm description of the implemented Algorithm 1 that discards the master routine in practice, in the revision.
>
> We thank the reviewer again for the valuable comments, and we appreciate any follow-up questions or potential reconsideration of the score.

---

> > ### Author Rebuttal · Reviewer_ctVz · 2026-04-02
> >
> > Thank you for the response. My concerns have been adequately addressed.

---

> > > ### Author Response · Authors · 2026-04-03
> > >
> > > Thank you for your reply and constructive questions. We will include these results in the revision for better practical relevance of our work.

---

### Official Review · Reviewer_DwAZ · 2026-03-13

**Soundness:** 2
**Presentation:** 3
**Significance:** 3
**Originality:** 3
**Overall Recommendation:** 4
**Confidence:** 2

**Summary:**

The paper studies online bidding in repeated second-price auctions (SPAs) when an advertiser’s true value is the marginal gain from paid exposure over organic outcomes, modeled as a treatment effect. It develops an online causal-learning algorithm that exploits SPA payments (the winner observes the highest other bid) to jointly estimate the treatment effect and optimize bids, achieving near-optimal regret under both full-information and binary feedback, with a matching lower bound. A key conceptual contribution is a provable separation from first-price auctions (FPAs), showing SPAs are fundamentally easier due to payment-revealed information.

**Compliance With Llm Reviewing Policy:**

Affirmed.

**Final Justification:**

I hve no more questions.

**Key Questions For Authors:**

- How sensitive is the regret to mis-specification of $(\omega, \lambda, \epsilon)$? Can these be data-driven, and what happens if λ is larger than anticipated?

- You allow HOB distributions with atoms. Could you elaborate on how large point masses affect the confidence widths $u_t(b)$ and the performance of Algorithm 2? Are there pathological cases near the optimal bid?

**Limitations:**

Yes.

**Strengths And Weaknesses:**

## Strengths

- This paper establishes a clear and rigorous separation between SPAs and FPAs under binary feedback by leveraging the payment-revealed HOB. Besides, the authors generalizes the IPW estimator analysis to allow arbitrary HOB estimation error envelopes $u_t(b)$, avoiding the stronger Bernstein-type conditions required in prior work and thereby broadening applicability.

- This paper provides matching upper and lower bounds and a complete minimax characterization under both feedback models; the theoretical development is cohesive and uses standard techniques appropriately.

## Weakness
- The locally bounded CDF assumption and the margin constraint $(1 − 4\epsilon − \lambda)/2 > 0$ are somewhat opaque; more intuition on their necessity and tightness would help, as would guidance on choosing \epsilon, especially given \lambda is unknown in practice.

- The bid discretization scales with $\sqrt{T}$, which raises computational and practicality concerns for long horizons; continuous optimization or adaptive discretization would be valuable.

---

> ### Author Rebuttal · Authors · 2026-03-30
>
> We appreciate the reviewer’s positive remark on the contributions of this study and questions about the current manuscript.
>
> **Example HOB:** The margin constraint $1-4\epsilon-\lambda>0$ is simply a guideline for choosing the constant $\epsilon\propto \frac{1-\lambda}{4}$. In practice, just like all algorithms with tuning parameters, we suggest to tune these parameters based on real data. For instance, a practical bidder can spend a few rounds to learn the HOB distribution and infer the parameters $(\omega,\lambda)$ (or infer them using the $O(\sqrt{T})$ initialization rounds). For clarity, we will replace the use of $\epsilon$ with an explicit factor $\frac{1-\lambda}{8}$ which is slightly relaxed to ensure the strict positive sign.
>
> **Bid discretization:** As the reviewer may have noticed, a $\sqrt{T}$-discretization is unnecessary. The reason for $\sqrt{T}$-discretization is that we compete against the *strongest* benchmark in the regret that picks the best bid from the *continuous interval*, which is never possible in practice. In practice, people often have a fixed, discrete set $B$ of bids of interest (of a reasonable size). Then it is natural to compete against an oracle benchmark that selects $b^*\in B$ within **this given set**. It is then straightforward to achieve the same regret guarantee as this work does, using this fixed $B$. In a word, this dense discretization arguably arises from the ambitious choice of benchmark. We also remark that the implementation only involves `np.max()`, `np.argmax()`, and `np.cumsum()` over the bid count array, which are fast.
>
> To comment on adaptive discretization, this is often possible in *non-contextual* setting, where one can continuously eliminate suboptimal bids and focus on less and less candidates. In the contextual case, every bid can be optimal from time to time. But once we have learned the value well, we can partition the value into multiple bins; for each bin it becomes non-contextual and we have a shrinking set of bids.
>
> **Mis-specification in HOB params:** This is an important question in practice. When the actual parameters are $(\omega\_0,\lambda\_0)$, with $\epsilon\propto \frac{1-\lambda}{4}$, the UCB computed from $(\omega_0,\lambda)$ is off by roughly $\propto \frac{1}{1-\lambda_0} - \frac{1}{1-\lambda}$, which naively leads to an additive term $O(T(\frac{1}{1-\lambda_0} - \frac{1}{1-\lambda}))$ in the regret. The sensitivity in $\omega$ can be understood in this same reasoning (since $\lambda$ can be viewed as a function of $\omega$): when we use the parameters $(\omega,\lambda)$ but the actual mass upper bound for this $\omega$ is $\lambda^*(\omega)>\lambda$, we reduce to the previous analysis.
>
> **Parameter estimation in practice:** There are in general two approaches to estimating $(\omega,\lambda)$. First and from a theoretical point of view, one can simply specify a family of HOB distributions of interest (such as truncated Gaussians); then the parameters can be computed accordingly. Second and most often in practice, people can use offline data to estimate an upper bound of $\lambda$ for some $\omega$ of interest, as the reviewer points out. The flexibility of choosing $\omega$ also makes this estimation relatively easier. We emphasize that, since an upper bound of true $\lambda$ suffices and the regret scales with $\frac{1}{1-\lambda}$, one can be very conservative as long as $1-\lambda$ is constant or polylog in $T$. Of course, being overly conservative hurts exploration rate in practice, so an alternative is to start with a conservative (estimated offline) $\lambda$ and decreases it progressively. The analysis of this alternative remains an interesting open problem.
>
> **Relation to UCB:** As commented above, the UCB and the final regret scales with roughly  $\frac{1}{\epsilon}\propto \frac{1}{1-\lambda}$. So the larger point mass allowed, the larger this leading constant is. As long as not all bids collapse, Definition 2.4 can be satisfied and the analysis remains valid.
>
> We thank the reviewer again for the valuable comments, and we appreciate any follow-up questions or potential reconsideration of the score. We also refer the reviewer to our response to Reviewer ctVz for some numerical experiments we will include in the revision.

---

> > ### Author Rebuttal · Reviewer_DwAZ · 2026-04-03
> >
> > I have no additional questions, and will raise my score accordingly.

---

> > > ### Author Response · Authors · 2026-04-03
> > >
> > > Thank you for your positive feedback. We will be sure to include the mentioned changes in the revision.

---

### Official Review · Reviewer_4hRy · 2026-03-16

**Soundness:** 4
**Presentation:** 2
**Significance:** 3
**Originality:** 4
**Overall Recommendation:** 4
**Confidence:** 3

**Summary:**

The paper studies online learning for bidding in repeated second-price auctions (SPA). In each auction, a winning bidder can observe the second highest bid and his true value. The paper proposes an interesting model, where a losing bidder can also get some value from the click of the organic result, and the treatment effect (i.e. the difference between winning value and losing value) can be modeled by the dot product of an unknown parameter vector ($\theta_*$) and a per-round feature vector $x_t$. The paper proves a $\Theta(\sqrt{dT})$ regret, where $d$ is the feature dimension.

**Compliance With Llm Reviewing Policy:**

Affirmed.

**Final Justification:**

As I get more idea from the author's explanation, I would raise my score.

**Key Questions For Authors:**

Can you explain the proof roadmap of the paper?

**Limitations:**

Yes

**Strengths And Weaknesses:**

Soundness:

I don’t think the paper has a major problem. I leave two minor comments here.

Line 242. The $M_t$ in formula (5) should be $m_t$.

Line 283 (&122). The integration by parts argument for $\bar{r}_{t,0}(b)$ seems to be completely wrong. How does this influence the proof? I would assume it does not matter but you should clarify this.

Originality and Significance:

The model where the losing bidder also gains value from organic is interesting and definitely worth studying in theory.

Presentation:

I feel that the paper’s organization needs significant improvement to highlight the proof roadmap. While the intro is very clear about the contribution of the paper, it is hard to understand what each part of the paper is doing in the later sections, especially given the highly technical nature of the paper and the dense double column ICML format.

For example, what message does Section 3 want to convey? The section does not have a clear goal, and the idea of learning reserve price distribution has already been studied in the bandit / online pricing / online auction literature. Section 4 talks about learning the treatment effect but each part is really messy.

Also I believe the authors should explain more about why SPA has a better regret compared to FPA.

---

> ### Author Rebuttal · Authors · 2026-03-30
>
> We appreciate the reviewer’s positive remark on the value of this study and pointing out a few typos and raising questions about the current manuscript.
>
> **Typos:** We have fixed Line 242. For Line 283, as the reviewer points out, the payment is a typo. The actual payment from integral by parts is as written in Line 227 above Lemma 3.1. This genuine typo does not affect the proofs, as the payment quantity is correctly handled everywhere else (for example, in Lemma 3.1).
>
> **Presentation:** We appreciate the reviewer’s question on the organization of the presentation. To clarify, the flow of this work is as follows:
> In Section 3, we aim to elaborate on why the second-price payment rule reveals more information and induces a one-sided feedback. While we agree that the HOB estimation under one-sided feedback has been studied in the literature (such as Han et al. 2025 mentioned in Line 212), this is included for completeness and to provide the intuition for the UCB in later sections.
> In Section 4, we incorporate the idea of treatment value estimation in SPAs to develop the bidding algorithm and the theoretical results. Since the target value ($v_{t,1}-v_{t,0}$) is never observable, we resort to the IPW estimator as an unbiased surrogate in Section 4.1, whose performance is summarized in Lemma 4.3.
> Nonetheless, the error bound in Lemma 4.3 scales inversely with the variance of the IPW estimator, so standard analysis fails. Section 4.2 sketches the high-level intuition of why the regret *can* be bounded even under large variance via two reward re-formulations. Then Section 4.3 provides concrete algorithm and theoretical result for this argument.
> Section 4.4 introduces a somewhat complicated idea from the existing literature to handle a dependence issue that was deliberately ignored in the previous sections (for demonstration). While this is a purely theoretical device, existing bounds without it typically only guarantee $O(d\sqrt{T})$ instead of optimal $O(\sqrt{dT})$.
> We will include an Organization section and a brief description at the beginning of these subsections to improve clarity.
>
> **SPA vs FPA:** The reason why bidding in SPAs enjoys a better regret than in FPAs is, as we commented in the contributions and in the conclusion, that additional information can be extracted from the second-price payment. This was also the message we hoped to convey via Section 3 and Figure 1.
>
> We thank the reviewer again for the valuable comments, and we appreciate any follow-up questions or potential reconsideration of the score. We also refer the reviewer to our response to Reviewer ctVz for some numerical experiments we will include in the revision.

---

> > ### Author Rebuttal · Reviewer_4hRy · 2026-04-01
> >
> > I'd be glad to raise my score to 4 after reading the rebuttal.

---

> > > ### Author Response · Authors · 2026-04-03
> > >
> > > Thank you for your positive feedback. We will be sure to include the mentioned changes in the revision.

---

### Decision · Program_Chairs · 2026-04-30

**Decision:**

Accept (regular)

**Comment:**

An existing line of work addresses learning the value while bidding in a first price auction (FPA).  This paper extends it to a second price auction (SPA) with a number of technical innovations and insights.  The reviewers were appreciative of these theoretical results as the main strength of the paper and I agree with them.  They raised some concerns about presentation and motivation that were largely satisfied during the discussion process with some proposed additions and changes that I agree would be helpful and should be implemented.

That said, while all the reviewers were at least weakly positive about the paper, none though the results important enough to champion it.  From my own reading of the paper I appreciate the theoretical contributions but question the relevance to practice, so while the paper would be fine to accept I view it as a low priority.

In particular, a key part of the motivation is tightly linked to search auctions, particularly in the way the approach models the possibility that the advertiser may have positive utility even if they lose the auction because they may appear in the organic search results.  But the model is quite far from search auction practice in several key ways which suggest to me that this paper has very limited practical relevance without substantial further development.
 - Prior work on FPA has been motived by ad exchanges, which are more from the display advertising part of the ecosystem and have quite different implementations in practice.  But a number of the resulting modeling choices seem to have been directly imported despite not making sense in search ads contexts, and the motivation provided in the paper even still refers to ad exchanges.  A driving factor is that in ad exchanges you can participate in real-time bidding, so it makes sense to think about feedback and context per auction.  Search auctions are more centralized and lack this option.  Feedback is generally coarser (you may learn only aggregate performance), the available context information is much more limited, and you lack the ability to bid on a per-auction basis (instead setting up campaigns).  You can opt into letting Google or Bing optimize on a per-auction basis for you, but then they do not have the information limitations this paper studies since they are running the auction.
- While this paper studies the SPA, in practice Google and Bing both use the generalized second price auction (GSP), which adds complexity because (a) there are multiple winners, not a single one and (b) the auction is no longer truthful.  This is even before considering other factors such as budgets.
- One proposed motivation is that some searchers may not look at the ads, but this is already factored into the click prediction that Google et al. do.  A key detail that is being elided is that in practice payment is conditional on generating a click, so the idea that winning is bad if you don't get clicked  doesn't apply, at least not in this simple way.